# Dipolar-stabilized first and second-order antiskyrmions in ferrimagnetic multilayers

Michael Heigl [1✉], Sabri Koraltan [2], Marek Vaňatka [3], Robert Kraft [2], Claas Abert [2,4], Christoph Vogler[2], Anna Semisalova [5], Ping Che [6], Aladin Ullrich [1], Timo Schmidt[1], Julian Hintermayr[1], Dirk Grundler [6,7], Michael Farle [5], Michal Urbánek [3], Dieter Suess[2,4] & Manfred Albrecht [1]

Skyrmions and antiskyrmions are topologically protected spin structures with opposite vorticities. Particularly in coexisting phases, these two types of magnetic quasi-particles may show fascinating physics and potential for spintronic devices. While skyrmions are observed in a wide range of materials, until now antiskyrmions were exclusive to materials with $D_{2d}$ symmetry. In this work, we show first and second-order antiskyrmions stabilized by magnetic dipole–dipole interaction in Fe/Gd-based multilayers. We modify the magnetic properties of the multilayers by Ir insertion layers. Using Lorentz transmission electron microscopy imaging, we observe coexisting antiskyrmions, Bloch skyrmions, and type-2 bubbles and determine the range of material properties and magnetic fields where the different spin objects form and dissipate. We perform micromagnetic simulations to obtain more insight into the studied system and conclude that the reduction of saturation magnetization and uniaxial magnetic anisotropy leads to the existence of this zoo of different spin objects and that they are primarily stabilized by dipolar interaction.

[1] Institute of Physics, University of Augsburg, Augsburg, Germany. [2] Faculty of Physics, University of Vienna, Vienna, Austria. [3] CEITEC BUT, Brno University of Technology, Brno, Czech Republic. [4] Research Platform MMM Mathematics - Magnetism - Materials, University of Vienna, Vienna, Austria. [5] Center for Nanointegration and Faculty of Physics, University of Duisburg-Essen, Duisburg, Germany. [6] Laboratory of Nanoscale Magnetic Materials and Magnonics, Institute of Materials (IMX), École Polytechnique Fédérale de Lausanne (EPFL), Lausanne, Switzerland. [7] Institute of Microengineering (IMT), École Polytechnique Fédérale de Lausanne (EPFL), Lausanne, Switzerland. ✉email: michael.heigl@uni-a.de

Over the last few years, topological nontrivial magnetic quasiparticles have attracted a lot of attention. Especially, skyrmions have been investigated very actively over the last decade. These cylindrical-like magnetic domains were observed in bulk[1–5], thin films[6–10], and monolayer[3,11] systems. Many concepts to stabilize skyrmions exist, while the use of Dzyaloshinskii–Moriya interaction (DMI) being the most common. Here generally, the system's inversion symmetry is broken whereby the asymmetric DMI locally tilts the uniform magnetic state[1–4,6–8]. An alternative approach is the use of the competition between long-range dipolar interaction, ferromagnetic exchange, and magnetic anisotropy. In this case, materials can not only host topological trivial bubbles (called type-2 bubbles) but also topologically protected chiral magnetic bubbles (type-1), resembling magnetic Bloch skyrmions[12–17]. These chiral bubbles or dipolar-stabilized skyrmions were intensively investigated in Fe/Gd multilayers by Montoya et al.[16,18,19]. In order to simplify the distinction in this work, we call type-1 bubbles *skyrmions* and type-2 bubbles *bubbles*.

Another novel magnetic quasiparticle that gained a lot of interest are magnetic *antiskyrmions*. They carry a negative vorticity ($m$) instead of the positive $m$ of skyrmions. Thus, the topological charge ($N_{sk}$) of a skyrmion and antiskyrmion with the same polarity ($p$) is opposite to each other ($N_{sk} = p \cdot m$). These first-order antiskyrmions exhibit a twofold symmetry and consist of alternating Néel- and Bloch-type domain walls that confine inside the out-of-plane magnetic moments from the surrounding antiparallel moments. While there have been many predictions of antiskyrmions stabilized by magnetic dipolar interaction[20,21], interface DMI[22,23], and noncentrosymmetric $D_{2d}$ symmetry[20,24,25], until now, only the latter could be experimentally realized in Heusler compounds[26–29]. In dipolar-dominated systems, only local artificial antiskyrmions could be created and observed[30]. Antiskyrmions with an additional iteration of a Néel- and a Bloch-type wall are called second-order antiskyrmions. They have the vorticity $m = -2$ and show a threefold symmetry. While they have been theoretically predicted[31], they have not been observed experimentally yet. Antiskyrmions in themselves are of interest for spintronic devices, but especially magnetic phases with a multitude of topologically protected spin objects are predicted to show a variety of fascinating phenomena for possible skyrmion–antiskyrmion-based spintronics. These phenomena include different motion types of skyrmions and antiskyrmions[32], skyrmion–antiskyrmion liquids[33], rectangular lattices[34], phase separation[35], topological conversion by

their collision[20,21,32], and annihilation that emits propagating spin waves[21]. Furthermore, in the original racetrack concept, information is stored by the distance between skyrmions. Due to defects and thermal fluctuation, there is a significant source of error[36] making it difficult to use. To overcome this issue, binary information given by two different topologically protected spin objects was proposed[37–39]. Very recently, the coexistence of antiskyrmions and skyrmions was observed as well in $D_{2d}$ Heusler compounds[40,41]. In these crystalline bulk systems, the spin objects exist only in a geometrical deformed state and in specific planes of the crystal.

In this work, we report on the observation of dipolar-stabilized antiskyrmions in Fe/Gd-based multilayers (MLs). The MLs were modified by Ir insertion layers to reduce the magnetic moment and anisotropy. We show coexisting phases of first- and second-order antiskyrmions, Bloch skyrmions, and topological trivial bubbles. We evaluate their stability in the dependence on out-of-plane (oop) magnetic field and temperature. From the experimental results, additionally supported by micromagnetic simulations, we conclude that the reduced magnetic moment and anisotropy supports the formation of antiskyrmions via dipolar interaction. Moreover, the nucleation process of the antiskyrmions is analyzed experimentally and theoretically. Our findings show that antiskyrmions can be stabilized outside of bulk $D_{2d}$ crystals solely by dipolar interaction. Further, we observe second-order antiskyrmions. We hope that this easy-to-access range of different coexisting magnetic quasiparticles in thin films opens many possibilities for future magnetic quasi-particle interaction and dynamic experiments.

## Results

We prepared a series of $[Fe(0.35)/Ir(0.20)/Gd(0.40)]_{N_{Ir}}/[Fe(0.35)/Gd(0.40)]_{80-2N_{Ir}}/[Fe(0.35)/Ir(0.20)/Gd(0.40)]_{N_{Ir}}$ MLs (all thicknesses in nm) with different repetition numbers $N_{Ir}$ of the Ir-containing layer stacks ($N_{Ir} = 0, 2, 5, 10, 20, 40$), where $N_{Ir} = 0$ is equivalent to $[Fe(0.35)/Gd(0.40)]_{80}$ and $N_{Ir} = 40$ to $[Fe(0.35)/Ir(0.20)/Gd(0.40)]_{80}$. The magnetization of the ferrimagnetic MLs is Gd-dominant over all measured temperatures $T$, as shown by the magnetic moment versus magnetic field ($M-H$) hysteresis loops provided in the supplementary information (Supplementary Fig. 9).

Figure 1a displays an exemplary Lorentz transmission electron microscope (LTEM) image of the $[Fe/Ir/Gd]_2/[Fe/Gd]_{76}/[Fe/Ir/Gd]_2$

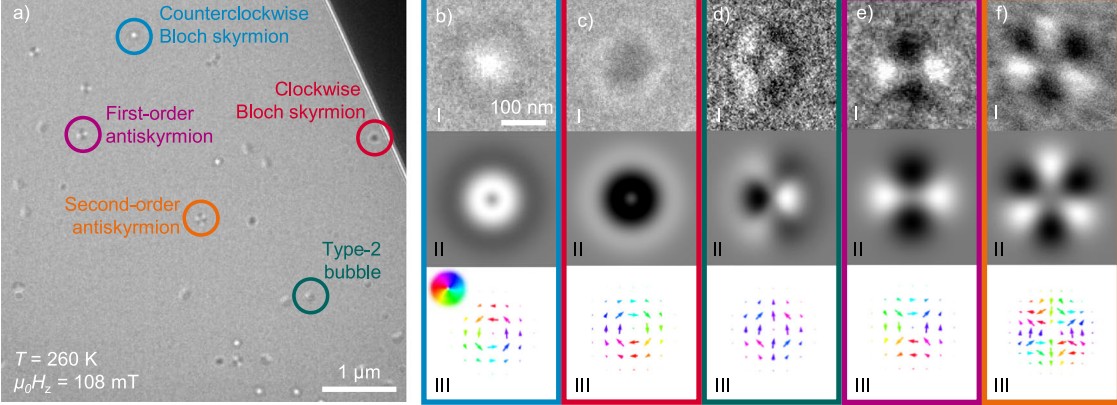

**Fig. 1 Variety of spin objects. a** Underfocused LTEM image of $[Fe/Ir/Gd]_2/[Fe/Gd]_{76}/[Fe/Ir/Gd]_2$ taken in an applied oop magnetic field $\mu_0 H_z$ of 108 mT at the temperature $T$ of 260 K. Five different coexisting spin objects are observed. The images **b**–**f**, (I) show the measured zoomed-in spin objects and **b**–**f**, (II) display the simulated LTEM contrast caused by the theoretical spin textures shown in **b**–**f**, (III). In **b**–**f**, III the arrows and their color depict the directions of the ip components, with their sizes displaying the strengths. **b** Bloch skyrmion with counter- and **c** with clockwise rotation, **d** type-2 bubble, **e** first-order antiskyrmion, and **f** second-order antiskyrmion.

sample taken at 260 K in an applied oop magnetic field $\mu_0 H_z$ of 108 mT. The image was acquired in 2.5-mm underfocus in Fresnel mode. Various spin objects are observed, as displayed in more detail in Fig. 1b–f (panel I) along with theoretical LTEM contrast images (panel II) and the corresponding spin textures (panel III), showing a counterclockwise (ccw) Bloch skyrmion (Fig. 1b), a clockwise (cw) Bloch skyrmion (Fig. 1c), a type-2 bubble (Fig. 1d), a first-order antiskyrmion (Fig. 1e), and a second-order antiskyrmion (Fig. 1f).

The theoretical contrast of the spin textures in Fig. 1b–f (panel II) was simulated by calculating the cross-product of the two-dimensional in-plane (ip) components of the spin objects (Fig. 1b–f, panel III) and the incident electron beam multiplied by a deflection factor. The resulting smoothed histograms are plotted in Fig. 1b–f (panel II). Comparing the experimental results with the theoretical ideal contrasts, we conclude that our samples exhibit up to five different spin objects at a time. While Bloch skyrmions and type-2 bubbles are well-known spin structures that have been previously observed in similar Fe/Gd ML systems[12–17], antiskyrmions have, to the best of our knowledge, never been observed experimentally outside of compounds with $D_{2d}$ symmetry. We also observe antiskyrmions of second order, which exhibit a threefold symmetry instead of the twofold symmetry of first-order antiskyrmions. The different spin objects have similar sizes of around 200–250 nm in diameter. Type-2 bubbles differentiate themselves from skyrmions by domain walls consisting of both Néel- and Bloch-type pointing roughly in one ip direction. Because of this, they also exhibit no rotational symmetry. The elliptical shape in Fig. 1d, I and II arises from the onefold symmetry of the spin configuration displayed in Fig. 1d, III. In addition, the second-order antiskyrmions appear to have the largest size due to the additional domain wall alteration between Bloch- and Néel-type. The different spin objects are randomly distributed over the whole imaged area revealing no preferred orientation and appear at different locations after resetting the magnetic state. Jiang et al.[42] showed that Néel skyrmions exhibit no LTEM contrast when the film is normal to the electron beam. Considering this, we tilted our film samples by up to 30° revealing no indication of the existence of pure Néel-type spin objects.

Using LTEM imaging and superconducting quantum interference device—vibrating sample magnetometry (SQUID-VSM),

we explore the dependence on applied oop magnetic field and temperature, which results in the formation and stabilization of different spin objects. Figure 2a shows the oop and ip $M$–$H$ hysteresis loops of $[Fe/Ir/Gd]_2/[Fe/Gd]_{76}/[Fe/Ir/Gd]_2$ at 250 K. Moreover, the corresponding LTEM images at different applied oop fields are presented in Fig. 2b, which were captured at the same location while sweeping the field from zero toward magnetic saturation. Both the oop and ip hysteresis loops saturate at similar fields and have a saturation magnetization of around 270 kA/m, although their courses differ greatly. The oop loop exhibits a linear growth of magnetization with only a small opening up to around 60 mT. This can be connected to the reversible continuous growth of stripes parallel to the field. At larger fields, the hysteresis loop opens because of irreversible processes, like the formation and annihilation of cylindrical spin textures[43]. In the LTEM images, we identify different ranges of stability of the different spin objects. Starting from zero field, the film exhibits two different kinds of stripe domains up to 63 mT: broader dark and bright stripes and narrower stripes with less contrast and half the periodicity of the broader stripes (Fig. 2b, I). While technical limitations of our experimental apparatus limit the analysis of such magnetic domains, they seem to be similar to domains seen in previous works[16,17] and described in detail in ref. [44]. The bordering in-plane domain wall components are aligned mostly parallel to the stripes. Antiparallel in-plane moments on the opposite sides of the stripe result in larger high-contrast stripes, while parallel aligned Bloch walls result in a narrower stripe pattern. In other words, the "broader" stripes exhibit domain walls with same chirality, while the "narrower" ones exhibit overall no chirality. However, the underlying periodicity of the magnetic stripes is about 250 nm and the same for both types. The existence of only one underlying periodicity was also confirmed by additional magnetic force microscopy measurements (Supplementary Fig. 12). The different chiralities of the stripes can determine the type of spin objects that can form under oop magnetic field. When an oop field is applied, domains with magnetization parallel to the field grow at the expense of stripes antiparallel to the field. At 63 mT, the first Bloch skyrmions start to form (Fig. 2b, II). We observe that they preferably arise from collapsed stripe domains with the same chirality. Equal numbers of counterclockwise (white) and clockwise (black) Bloch

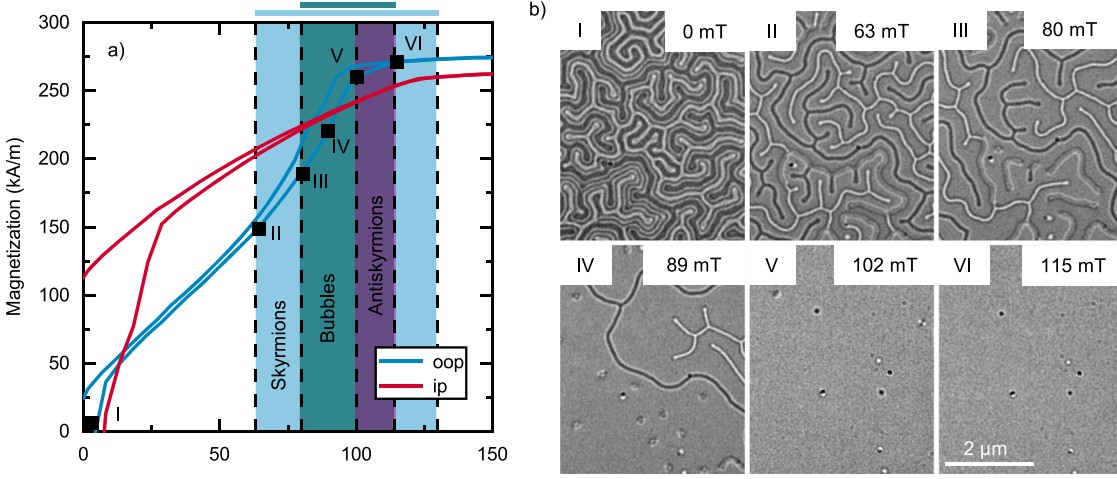

**Fig. 2 Stability ranges of the spin objects. a** Oop and ip $M$–$H$ hysteresis loops of $[Fe/Ir/Gd]_2/[Fe/Gd]_{76}/[Fe/Ir/Gd]_2$ at 250 K. The colored regions mark the stability ranges of the spin objects coming from zero field. **b** LTEM images of the sample at different oop magnetic fields that are also marked in the oop loop in (**a**). Panel I shows stripe domains with chirality (broader bright and dark stripes) and without (thinner stripes). At 63 mT, disordered stripes coexist with Bloch skyrmions (II). Between 80 and 89 mT, type-2 bubbles start to appear (III–IV). In panel V, all stripes disappeared and antiskyrmions nucleated at 102 mT. Antiskyrmions, skyrmions, and bubbles coexist in this field range. At 115 mT, only skyrmions are stable and observable (VI).

skyrmions appear when averaged over the whole sample. When we further increase the magnetic field to 80 mT, type-2 bubbles start to form out of the stripes without chirality (Fig. 2b, III). At 89 mT, the majority of stripes has vanished or shrank down to circular spin objects. The number of skyrmions stayed constant from 80 mT, while the number of type-2 bubbles has further increased (Fig. 2b, IV). The first isolated antiskyrmions appear at 102 mT when all the remaining stripes have vanished (Fig. 2b, V). The number of topologically unprotected bubbles is also strongly reduced at these higher fields. The last LTEM image was captured at 115 mT (Fig. 2b, VI). Only randomly distributed Bloch skyrmions are left. Their size decreased with increasing field down to around 100 nm and they dissipate at around 130 mT. All six panels also show microstructural defects as black dots that are slightly smaller than the skyrmions and unaffected by the magnetic field. In Fig. 2b, II–V, one of the counterclockwise skyrmion is pinned to one of these defects. We conclude that Bloch skyrmions are stable over the largest field range, while bubbles and antiskyrmions only exist in rather limited field ranges.

We repeated this procedure for different temperatures between 100 and 300 K. The observed stability ranges of the different spin objects of $[Fe/Ir/Gd]_2/[Fe/Gd]_{76}/[Fe/Ir/Gd]_2$ are displayed in Fig. 3. The diagram was constructed using the data from the LTEM measurements marked by the black dots. The area in-between the experimental data was filled under the assumption that stability transitions happen in the middle of the measured data points. As already mentioned, the domain morphology and position of the different spin objects seem to be randomly

distributed for every new field sweep starting from zero field after saturation. At the same time, the range of stability of the different spin objects stayed the same for every field sweep. Further, we did not observe a pure antiskyrmion phase. They always coexisted with Bloch skyrmions and sometimes with type-2 bubbles and/or stripes. It was also evident that the stability range of antiskyrmions is smaller than the one of skyrmions for both temperature and field. Second-order antiskyrmions were only observable at 260 K, but stayed stable for larger magnetic fields in comparison to first-order antiskyrmions.

We also measured and created phase diagrams of the five other samples of our series ($N_{Ir}$ = 0, 5, 10, 20, 40). The other phase diagrams (Supplementary Fig. 11), hysteresis loops (Supplementary Fig. 9), and additional LTEM images (Supplementary Fig. 13) are available in the Supplementary Information. Generally, more Ir insertion layers decrease both magnetization and uniaxial magnetic anisotropy (Supplementary Fig. 10). Because of a too low magnetization, it was not possible to image the sample with $N_{Ir}$ = 40 by LTEM. The other five samples showed antiskyrmions at room temperature. While samples with $N_{Ir}$ = 0, 2, 10, and 20 only show antiskyrmions at larger temperatures, the sample with $N_{Ir}$ = 5 exhibits antiskyrmions at all measured temperatures. First-order antiskyrmions were observed at a wide range of temperatures in every sample of our series besides $N_{Ir}$ = 40; however, second-order antiskyrmions were only observed at 260 K in $N_{Ir}$ = 2 and at 300 K in $N_{Ir}$ = 20. In both cases, they were quite rare in comparison to the other spin objects. Because of that, we cannot rule out their existence at other temperatures and fields, but conclude that their nucleation process is less likely in our samples in comparison to first-order antiskyrmions.

Micromagnetic simulations were performed based on the finite difference method where the magnetization dynamics was investigated by means of numerical integration of the Landau–Lifshitz–Gilbert (LLG) equation to reproduce our experimental findings. Here, we considered only the magnetic exchange, uniaxial magnetic anisotropy, demagnetization, and Zeeman fields, using an exchange stiffness constant $A_{ex}$ = 6 pJ/m², a uniaxial magnetic anisotropy constant $K_u$ = 22.35 kJ/m³, a saturation magnetization $M_s$ = 225 kA/m, and a film thickness of 62 nm as material parameters, which is in the range of our experimental values (Supplementary Fig. 10). Details about the simulation method can be found in the "Methods" section. In order to confirm the dipole–dipole interaction as stabilization mechanism for antiskyrmions, no DMI was included in the simulations. The results are displayed in Fig. 4. The simulations did not show significant differences in their spin configuration along the film thickness. Figure 4a–d shows the domain morphology at a larger scale with the color illustrating the z-component of the magnetization. Figure 4e–g displays an enlarged portion with black vectors showing the x- and y-components of the magnetization. Figure 4a and e displays the relaxed magnetization state at zero field showing stripes with different chiralities bristled with Bloch lines. In Fig. 4b, an external oop magnetic field of $\mu_0 H_z$ = 50 mT is applied and the stripes start to shrink down forming skyrmions. At an external magnetic field of $\mu_0 H_z$ = 88 mT the coexistence of type-2 bubbles is revealed, Bloch skyrmions with both cw and ccw chiralities, and antiskyrmions (Fig. 4c, f). In Fig. 4d and g, only topologically protected spin structures (skyrmions/antiskyrmions) remain at 95 mT. The micromagnetic simulations are in full agreement with our experiments (Fig. 2b). They exhibit the same coexistence of bubbles, skyrmions, and antiskyrmions with a decreasing density of bubbles for larger applied fields.

To confirm the metastable state of antiskyrmions in our system, micromagnetic simulations of the energy barrier using a hybrid finite- and boundary element method were performed.

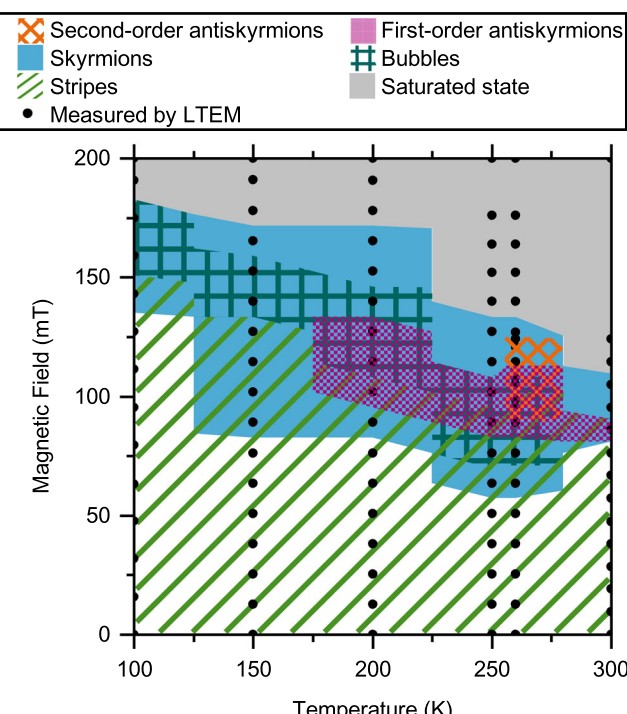

**Fig. 3 Phase map of the spin objects.** Magnetic field and temperature dependence of the stability ranges of the different spin objects of [Fe/Ir/Gd]$_2$/[Fe/Gd]$_{76}$/[Fe/Ir/Gd]$_2$. The diagram was constructed using the data from the LTEM measurements (black points). A pure antiskyrmion phase (purple checkerboard pattern) is not observable. They always coexist with Bloch skyrmions (blue) and sometimes with stripes and/or bubbles (green stripes). The stability range of antiskyrmions is also smaller than the one of skyrmions for both temperature and field. Second-order antiskyrmions were only observable at 260 K, but stayed stable for larger magnetic fields in comparison to first-order antiskyrmions.

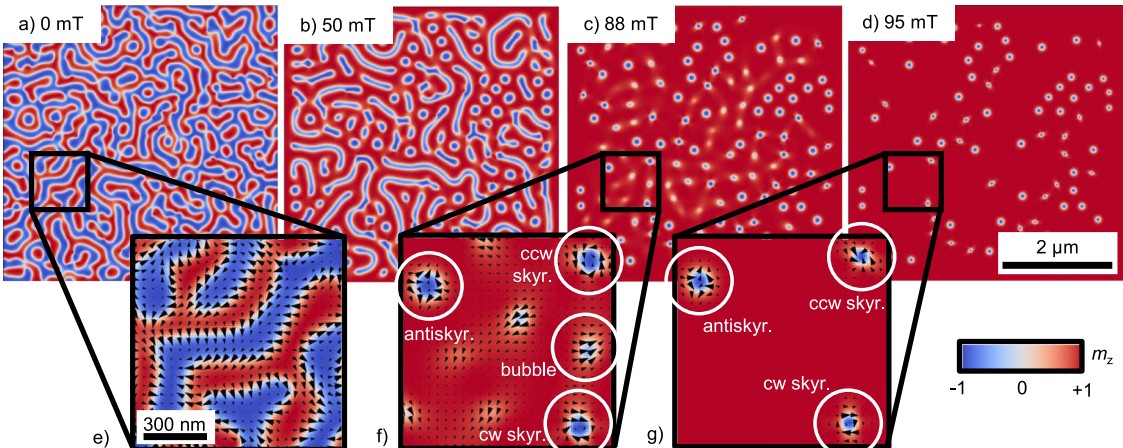

**Fig. 4 Micromagnetic simulations of the domain morphology.** The domain morphology is simulated using magnetic parameters close to the sample [Fe/Ir/Gd]$_2$/[Fe/Gd]$_{76}$/[Fe/Ir/Gd]$_2$. **a–d** show the domain morphology at a larger scale with the color illustrating the $z$-component of the magnetization. **e–g** display an enlarged portion with black vectors showing the $x$- and $y$-components of the magnetization. **a, e** display the relaxed magnetization state at zero field showing stripes with different chiralities bristled with Bloch lines. In (**b**), an external magnetic field of $\mu_0 H_z = 50$ mT is applied, stripes and skyrmions are coexisting. An external magnetic field of $\mu_0 H_z = 88$ mT reveals the coexistence of type-2 bubbles, Bloch skyrmions with both cw and ccw chiralities, and antiskyrmions (**c, f**). In (**d, g**), only topologically protected spin structures (skyrmions/antiskyrmions) remain at 95 mT.

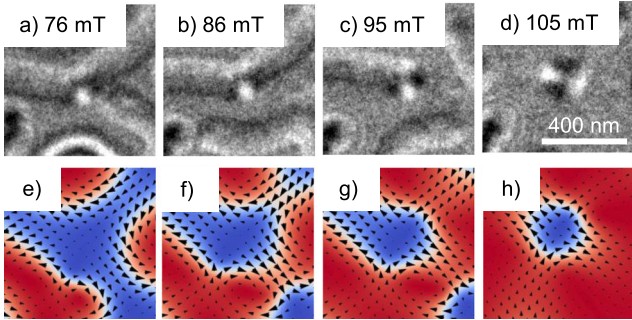

**Fig. 5 Nucleation process of antiskyrmions. a–d** Nucleation process of an antiskyrmion starting from a crossing point of three stripes without chirality (**a**) imaged by LTEM. **b, c** show the crossing point with only two and one stripe connected after the field was increased to 86 and 95 mT, respectively. The crossing point resembles a Bloch line and acts as the nucleation point of the antiskyrmion after the final stripe shrank down at 105 mT (**d**). As a comparison, the nucleation process in the micromagnetic simulations is displayed in (**e–h**). The simulation parameters are the same as in Fig. 4. A very similar process is observed here.

They showed a clear energy barrier between the antiskyrmion state and the skyrmion/saturated state (Supplementary Fig. 19). Thus, we conclude that the observed antiskyrmions are metastable and primarily stabilized by dipole–dipole interaction.

To learn how to achieve this state, we took a closer look at the nucleation process of an antiskyrmion both with LTEM imaging and micromagnetic simulations. Figure 5a–d shows a specific location of the [Fe/Ir/Gd]$_2$/[Fe/Gd]$_{76}$/[Fe/Ir/Gd]$_2$ sample at 200 K where an antiskyrmion eventually forms. We observed the distinct feature that in all samples at $T < 250$ K, antiskyrmions always nucleated from a crossing point of three stripes. One of these crossing points is displayed in Fig. 5a. Three stripes without a chirality meet, and due to the resulting in-plane components, one stable Bloch line emerges, resulting in a contrast very similar to an antiskyrmion. With increasing field, the stripes shrink down and one of them disconnects at 86 mT (Fig. 5b). At 95 mT, the Bloch line remains only connected to the very end of a single stripe. This stripe decreases in size until an isolated antiskyrmion

is left at 105 mT (Fig. 5d). It is important to note that these crossing points of three stripes also exist in samples that do not exhibit antiskyrmions. Also, not all triple crossings of stripes with Bloch lines necessarily nucleate antiskyrmions, if antiskyrmions exist. While the starting chirality of the crossing stripes does not seem to play a role, with increasing fields, the antiskyrmion always nucleated at the end of a stripe without chirality resembling a Bloch line. Very similar nucleation processes can be found in the micromagnetic simulation, as displayed in Fig. 5e–h. The process also starts at a crossing point of three stripes (Fig. 5e) and ends with an isolated antiskyrmion (Fig. 5h). In our simulation, antiskyrmions exclusively originate from these triple-crossing points for saturation magnetization values larger than 175 kA/m using a $K_u$ value of 22.35 kJ/m$^3$. This behavior agrees well with our experimental data. Additional nucleation processes of the other spin objects are displayed in the supplementary information (Supplementary Figs. 14 and 15).

To get a better understanding of the necessary parameter space to stabilize the different spin objects, the relevant magnetic properties were measured. We extracted $M_s$ from magnetometry measurements, while ferromagnetic resonance (FMR) measurements were carried out to determine the $K_u$ values. Due to the inclusion of heavy metal materials in the form of Ir in our MLs, an interface DMI contribution has to be considered. It was also shown that bulk DMI can even exist in inhomogeneous ferrimagnetic alloys[45]. We measured the DMI constant $D$ by Brillouin light scattering (BLS)[46]. The Fe/Gd ML without Ir showed no measurable DMI, while the ML with five Ir layers at the top and bottom ($N_{Ir} = 5$) exhibited a $D$ value of $(0.10 \pm 0.01)$ mJ/m$^2$. This value is more than an order of magnitude smaller than typical values for DMI-stabilized skyrmions[47]. Previous works[16,18] established that the ratio of $K_u$ and $M_s$ is crucial for the formation of stripe domains and skyrmions[48,49]. In Fig. 6, we plot the $M_s$ and $K_u$ values of our samples obtained for temperatures between 100 and 300 K, and mark the corresponding topologically protected spin objects that were experimentally observed. In addition, the results reported by Montoya et al.[16] obtained for similar systems are included exhibiting exclusively Bloch skyrmions at higher $M_s$ values (blue triangles). The purple squares correspond to samples that showed antiskyrmions under the investigated applied oop magnetic fields. These antiskyrmions always

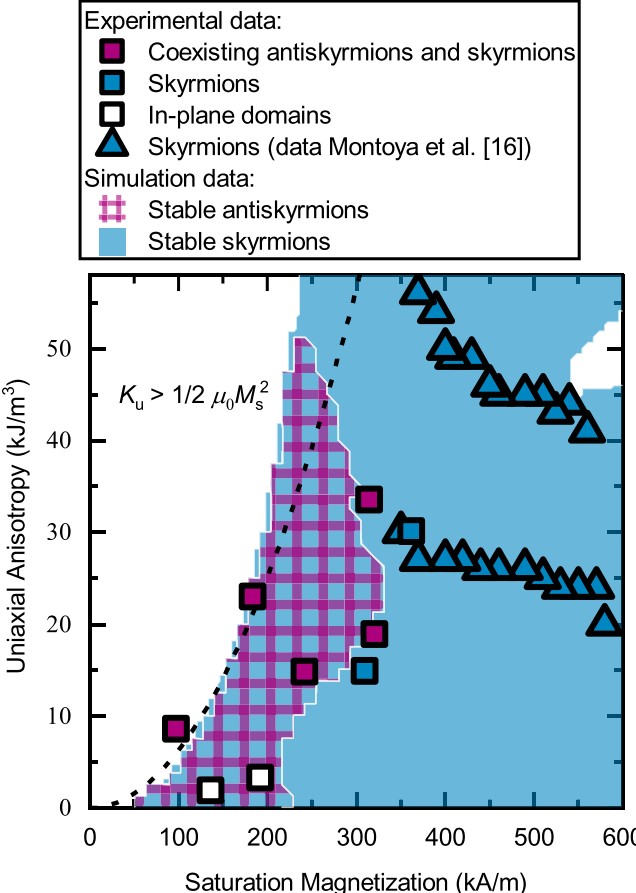

**Fig. 6 Phase diagram of antiskyrmions and skyrmions.** The diagram includes both experimental and simulation results, depending on the uniaxial magnetic anisotropy and saturation magnetization. The squares correspond to LTEM measurements that show antiskyrmions and skyrmions coexisting (purple), solely skyrmions (blue), and no visible spin objects (white) and their $K_u$ and $M_s$ values at the measured temperature. The triangles are experimental data points reported by Montoya et al.[16] obtained for similar Fe/Gd systems that show exclusively skyrmions. The colored areas of the diagram correspond to the parameter regions in which skyrmions (blue) and antiskyrmions (purple) are stable in our simulations. In the white regions, neither of them are stable. The dashed line shows $K_u = \frac{1}{2}\mu_0 M_s^2$.

coexisted with skyrmions and sometimes with bubbles. Blue and white squares correspond to the existence of sole skyrmions and in-plane domains without further spin objects, respectively. Figure 6 reveals the importance of low saturation magnetization values satisfying the condition $\frac{1}{2}\mu_0 M_s^2 > K_u$ for stabilizing antiskyrmions.

In order to get an understanding of how $M_s$ and $K_u$ affect the underlying spin texture, additional micromagnetic simulations were performed. The simulations were carried out with zero DMI and a constant film thickness of 62 nm. For different thicknesses and DMI values, additional simulations were conducted, showing only a minor impact on the formation and stability of the spin objects in the analyzed parameter range (Supplementary Fig. 18). However, for DMI values greater than 0.2 mJ/m², antiskyrmions do no longer exist and skyrmions dominate. The starting spin configuration for the simulations was an isolated Bloch skyrmion and an isolated antiskyrmion. Their stability was probed at different applied oop fields between 0 and 125 mT. The regions where these spin objects are at least stable at one of the fields are marked by color in Fig. 6. The purple region marks the parameter

range in which the initial antiskyrmion stays stable. The blue region marks parameters in which the skyrmion is stable. The underlying diagrams for specific fields and initial states are displayed in the supplementary information (Supplementary Figs. 16 and 17). It is important to note that the simulations do not convey the possible formation path of (anti-)skyrmions. This leads to stable skyrmions and antiskyrmions at very low $K_u$ values, while experimentally this parameter range leads to pure in-plane domains instead of stripe domains that are necessary to form the described spin objects (white squares). The simulations reveal a pocket in the phase diagram at small $K_u$ and $M_s$ where antiskyrmions are at least metastable. In agreement with our experiments, there is no exclusive antiskyrmion phase. The simulations also show that various spin objects exist if the magnetic shape anisotropy ($\frac{1}{2}\mu_0 M_s^2$) is larger than $K_u$ (right side of the dashed line). The simulation matches our experimental data very well, revealing a delicate balance of interactions that we achieved for our samples by the insertion of Ir layers moving our samples' parameters further to the novel phase region of metastable antiskyrmions.

In conclusion, we demonstrated the possibility of stabilizing antiskyrmions primarily by dipole–dipole interaction at room temperature. With LTEM imaging, we observe second-order antiskyrmions experimentally and first-order antiskyrmions outside of $D_{2d}$ Heusler compounds. The novel spin objects coexist in Fe/Gd-based multilayers together with Bloch skyrmions and topologically trivial type-2 bubbles. Phase diagrams of the spin objects were created in dependence on magnetic oop field, temperature, saturation magnetization, and uniaxial magnetic anisotropy. Micromagnetic simulations confirmed the phase pocket of metastable antiskyrmions for low saturation magnetization and for uniaxial magnetic anisotropy values satisfying the condition $\frac{1}{2}\mu_0 M_s^2 > K_u$. Further, we investigated the nucleation process of antiskyrmions and revealed the necessity of a crossing point of three magnetic stripe domains to form an isolated antiskyrmion with an oop magnetic field. This discovery significantly simplifies future investigations of antiskyrmions. In addition, the coexisting phases of different topologically protected spin objects provide great potential for further studies on quasi-particle interactions, spin dynamics, as well as for possible future applications in spintronics.

## Methods

**Sample preparation and characterization.** The ML samples were prepared at room temperature by dc magnetron sputtering from elemental targets on 30-nm-thick SiN membranes for LTEM imaging and on Si(001) substrates with a 100-nm-thick thermally oxidized $SiO_x$ layer for magnetic characterization. The sputter process was carried out with an Ar working pressure of $5 \times 10^{-3}$ mbar in an ultra-high-vacuum chamber (base pressure $<10^{-8}$ mbar). First, a series of [Fe(0.35)/Gd $(t_{Gd})]_{80}$ (all thicknesses in nm) was prepared with the Gd thickness $t_{Gd}$ ranging from 0.35 to 0.50 nm. For all samples, 5-nm-thick Pt seed and cover layers were used to protect the films from corrosion. The thicknesses of the layers were estimated from the areal densities measured by a quartz balance before deposition. It is important to note that the deposition of layers in the sub-0.1-nm range is not precisely possible with this technique. Reproducibility can be only accomplished within one sputter run. It was previously reported that the structure of the MLs exhibits intermixing[16].

A composition that exhibits skyrmions at room temperature was achieved for $t_{Gd} = 0.40$ nm and was further modified by inserting 0.20-nm-thick Ir layers from the top and bottom. Samples with different numbers of insertion layers ($N_{Ir} = 0, 2, 5, 10, 20, 40$) were prepared. A schematic image of the [Fe(0.35)/Ir(0.20)/Gd $(0.40)]_{N_{Ir}}$/[Fe(0.35)/Gd(0.40)$]_{80-2N_{Ir}}$/[Fe(0.35)/Ir(0.20)/Gd(0.40)$]_{N_{Ir}}$ layer stack is displayed in Fig. 7.

The MLs were investigated by a variety of techniques at temperatures between 50 and 350 K. The integral magnetic properties of the MLs were probed by superconducting quantum interference device—vibrating sample magnetometry (SQUID-VSM). $M$–$H$ hysteresis loops were measured in oop and ip configuration for all samples at 50, 100, 150, 200, 250, 300, and 350 K (Supplementary Fig. 9) and the $M_s$ values were extracted.

The uniaxial magnetic anisotropy constant $K_u$ was evaluated using ferromagnetic resonance (FMR) measurements performed with a Bruker spectrometer and a

conventional X-band resonator at a microwave frequency $f = 9.45$ GHz. FMR spectra were recorded in the temperature range 100–300 K for different orientations of the static magnetic field with respect to the Fe/Gd ML plane (polar angular dependence). The $g$-factor was extracted from frequency-dependent FMR measurements at room temperature with a coplanar waveguide and microwave generator. The angular-dependent resonance field was analyzed within the Smit–Beljers approach that yields the value of the effective magnetization. One extracts the uniaxial magnetic anisotropy constant $K_u$ using the known value of the saturation magnetization.

The interface Dzyaloshinskii–Moriya interaction (DMI) was extracted from nonreciprocal spin wave dispersion using Brillouin light scattering (BLS) measurements (Fig. 8). The wave-vector-resolved BLS in backscattering geometry[50] was conducted at room temperature using a monochromatic continuous-wave solid-state laser with a wavelength of 473 nm. The incident angle was $\theta$ with respect to the normal axis of the film ($z$-axis). The external field $H$ was applied along the $y$ axis. Backscattered light was collected and processed with a six-pass Fabry–Perot interferometer TFP-2 (JRS Scientific Instruments). In this configuration, magnetostatic surface spin waves (MSSW)[51] with wave vectors $k_{stokes}$ and $k_{antistokes}$ were excited in the MLs. By varying the incident angle, the wave vectors were tuned. MSSW propagating along opposite directions led to resonance peaks of Stokes and anti-Stokes signals in the BLS spectra. The two peaks are asymmetric in both peak intensity and frequency. The intensity difference is attributed to the nonreciprocity of MSSWs. The frequency difference $\Delta f$ originates from the asymmetry of the dispersion relations resulting from the interface DMI. $\Delta f$ is used to determine the interface DMI constant $D$ according to ref. [46]. The linear fitting of this function was made at four different external fields and an average value of the interface DMI constant of $(0.10 \pm 0.01)$ mJ/m² was extracted. Error bars indicate the standard deviation of the four fitted values.

The magnetic domain morphology was imaged by Lorentz TEM using a FEI Titan and a JEOL NEOARM-200F system with a defocus of –2.5 mm in Fresnel mode. The temperature was controlled by the Gatan Double Tilt Liquid Nitrogen Cooling Holder Model 636 for the low-temperature measurements. The theoretical contrast of the spin textures in Fig. 1b–f (panel II) was simulated in the following way: parallel incoming electrons are modeled as vectors to impinge on the spin

structure on a discrete 2D grid at an angle of 90°. Analytical functions of the ip component of the 2D spin objects are used to model the internal magnetic field as vectors. In the magnetic field of the structure, electrons are deflected by the Lorentz force into a direction perpendicular to their own velocity and the local magnetization direction of the spin structure. In the code, the cross-product of the magnetic field vectors and the incoming electron vectors is multiplied by a scaling factor to adjust for weaker or stronger deflection. After iterating over all electrons/vectors in the image and calculating their impact points on the virtual detector, the collected data are converted to a continuous probability density using the kdeplot function from the seaborn Python library, implementing a kernel density estimation. The result is a continuous grayscale image, where bright areas indicate areas toward which many electrons/vectors are deflected, whereas dark regions indicate regions with a below-average electron/vector incident count. The program additionally plots a representation of the spin structure as a discrete grid of arrows, whose length and direction indicate the local orientation and magnitude of the spin structure. The arrows' colors are mapped to their direction on a HSV spectrum using the arctan2 function from the Python library numpy and are shown in Fig. 1b–f (panel III).

**Micromagnetic modeling**. The micromagnetic simulations were performed by using simulation codes based on both finite difference method, `magnum.af`[52] and hybrid finite- and boundary element method, `magnum.fe`[53]. While the former was used to reproduce the experimental results and investigate the formation process of experimentally observed spin structures, the latter was employed to investigate the stability of isolated skyrmions and antiskyrmions by varying the material parameters of the system. In both cases, the magnetization dynamics was investigated by means of numerical integration of the Landau–Lifshitz–Gilbert (LLG) equation[54–56],

$$\frac{\partial m}{\partial t} = -\frac{\gamma}{1 + \alpha^2} m \times H^{\text{eff}} - \frac{\alpha\gamma}{1 + \alpha^2} m \times \left(m \times H^{\text{eff}}\right), \quad (1)$$

where $\alpha$ is the Gilbert damping constant, $\gamma$ the reduced gyromagnetic ratio, $m$ the magnetization unit vector, and $H^{\text{eff}}$ is the effective field term, which includes the considered energy contributions.

To reproduce the field-driven magnetization dynamics in a Fe/Gd multilayer, we discretize a continuous film with a length $l = 5$ μm, width $w = 5$ μm, and a thickness $t = 62$ nm in `magnum.af`. The cell volume is chosen according to the exchange length $l_{ex} = 13$ nm as $l_x \times l_y \times l_z = 10 \times 10 \times 8.85$ nm³. First, each cell is randomly magnetized. We relax the structure to its ground state at vanishing external fields, where we model the effective field term $H^{\text{eff}}$ in such a fashion that it includes only the micromagnetic exchange, uniaxial magnetic anisotropy, and demagnetization fields, where the exchange stiffness constant $A_{ex} = 6$ pJ/m², uniaxial magnetic anisotropy constant $K_u = 22.35$ kJ/m³, and saturation magnetization $M_s = 225$ kA/m are used as material parameters. To investigate the dynamics of the system, the sample is subject to a time-dependent Zeeman field, where $H_{zee} = H(t)e_z$, with $e_z$ being the unit vector in the oop direction. $\mu_0 H(t)$ is chosen to increase linearly from 0 mT to 200 mT over 40 ns. The role of DMI was investigated by repeating the simulations, where DMI energy was modeled into the effective field term using $D = 0.10$ mJ/m².

In order to investigate the stability of an isolated spin object, we use `magnum.fe`. Here, we use a rectangular finite element mesh of dimensions $l = 300$ nm, $w = 300$ nm, and $t = 62$ nm. We chose simple parameterizations of the (anti-)skyrmions as initial magnetization states and solve the LLG for 10 ns at high damping $\alpha = 1$ in the presence of Zeeman fields with magnitudes between 0 and 125 mT. To reproduce the effect of isolation in an infinite film, we set a very large oop uniaxial magnetic anisotropy constant in the outer region of a circle in the $xy$-plane with radius $r = 150$ nm, well above the (anti-)skyrmion size, where $K_u = 1$ MJ/m³. We vary $M_s$ and $K_u$ and calculate the integer topological charge $N_{sk}$ of the

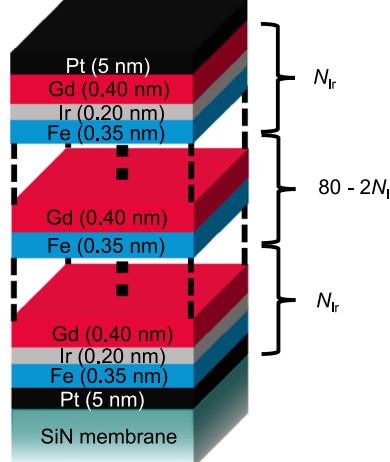

**Fig. 7 Schematic layer stack.** Schematic image illustrating the layer stack of the different Fe/Gd-based multilayers with Ir insertion layers.

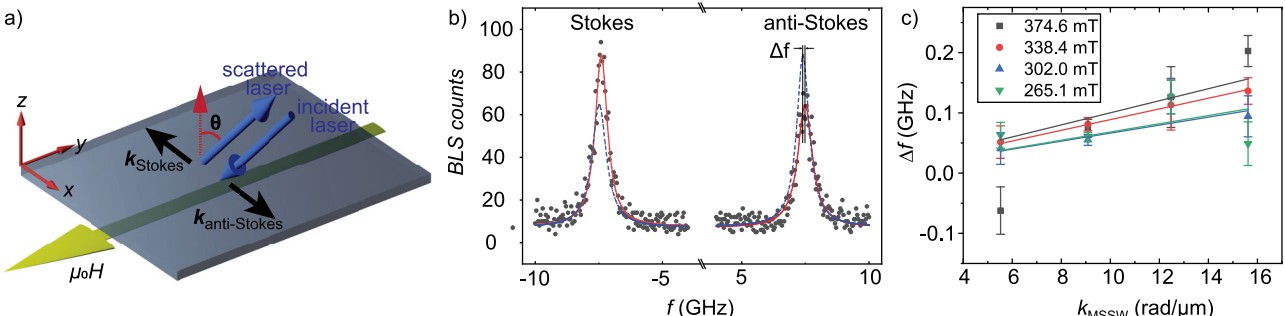

**Fig. 8 Brillouin light scattering measurements. a** Schematic diagram of backscattering geometry of the BLS measurement. The magnetic field was applied along the $y$ axis. Data were taken in backscattering geometry with different angles $\theta$ with respect to the $z$ axis. **b** BLS spectra of magnetostatic surface spin waves (MSSW) when $\mu_0 H = 338.4$ mT at $\theta = 20°$. Gray dots are the experimental data collected by BLS. Red lines are fitted curves with Lorentz function. Blue dash lines are the mirror curves of the red lines. $\Delta f$ is the frequency difference between Stokes and anti-Stokes peaks. **c** Frequency differences $\Delta f$ obtained at different field values. Solid lines are linear fits.

final magnetization states, defined by

$$N_{sk} = \int \frac{1}{4\pi} m \cdot \left( \frac{\partial m}{\partial x} \times \frac{\partial m}{\partial y} \right) dx dy. \tag{2}$$

If $N_{sk}$ of the relaxed magnetization state is $N_{sk} = -1$, an antiskyrmion is stable for the particular pair of $M_s$ and $K_u$, while a skyrmion is assumed to be stable for $N_{sk} = +1$. The exchange stiffness constant is chosen constant at $A_{ex} = 6$ pJ/m$^2$.

## Data availability

The datasets generated during and/or analyzed during this study are available from the corresponding author on reasonable request.

## Code availability

The Python code used to simulate the spin object contrasts is available under[57]. The micromagnetic simulations were performed using the closed source code of magnum.af[52] and magnum.fe[53].

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

## Acknowledgements

The funding of this project by the German Research Foundation (DFG) within the Transregional Collaborative Research Center TRR 80 "From electronic correlations to functionality", by the Bavarian-Czech Academic Agency (project no. 8E18B051), the FWF via the project FWF I-4917, and SNSF via Sinergia project 171003 (Nanoskyrmionics) is gratefully acknowledged. CzechNanoLab project LM2018110 funded by MEYS CR is gratefully acknowledged for the financial support of the LTEM measurements at CEITEC Nano Research Infrastructure. The computational results presented have been achieved, in part, using the Vienna Scientific Cluster (VSC). We thank Ms. Babli Bhagat (UDE) for the help with FMR measurements.

## Author contributions

M.A. and M.H. designed the experiments. M.H. prepared the samples and performed the magnetic characterization by SQUID-VSM. S.K., R.K., C.V., D.S., and C.A. performed the micromagnetic simulations. M.V., A.U., J.H., M.U., and M.H. carried out the LTEM measurements and discussed and analyzed the experimental data. A.S. and M.F. measured and analyzed the FMR results. P.C. and D.G. executed and analyzed the DMI measurements. T.S. performed SQUID-VSM measurements and MFM precharacterization. All authors contributed in writing and reviewing the paper.

## Funding

## Competing interests

The authors declare no competing interests.
