## [Peer Review File · Nature Communications]

Reviewers' Comments:

Reviewer #1:

Remarks to the Author:

I have reviewed manuscript NCOMMS-20-43292-T by M. Heigl and colleagues in which they report the first experimental demonstration of dipole-stabilized first and second-order antiskyrmions in thin-film multilayers. The manuscript presents a combination of experimental and numerical studies that show distinct topological spin textures (skyrmions, bubbles, antiskyrmions) can simultaneously form in ferrimagnetic Fe/Gd-based multilayers with/without heavy metal Ir thin-film layers. Using Lorentz TEM, the domain states were imaged under different magnetic field and temperature conditions which revealed field-induced evolution of stripes into distinct topological spin textures, as well as, insights on the field-stability of skyrmions ($S=-1,0,1$). Magnetic characterization of the six composition Fe/Gd-based multilayers showed that first-order and second-order antiskyrmions become favorable in materials that possess low uniaxial anisotropy and low magnetization. Furthermore, micromagnetic simulations revealed similar coexisting spin textures (skyrmion, bubbles, antiskyrmions) can form solely via dipolar-fields in continuous thick material slabs; simulations also revealed that interfacial DMI could be detrimental to the formation of antiskyrmions. In general, these results provide a potential guide to further design new materials that can host controllable antiskyrmion phases in thin-film multilayers, a subject of broad interest to academic and industry researchers investigating potential exploitation of chiral states in spintronic-based technologies.

After careful review, I believe the novelty and quality of the results presented in the manuscript can merit publication in Nature Communications after the authors have addressed comments/questions appended below. Revisions to the overall syntax are suggested to better communicate the presented results, as well as, expand the subject matter in the main manuscript. Since Nature Communications facilitates publishing works with up to 5000 words (not including abstract, methods, references and figure legends), I suggest the authors bring results from the supplementary into the main manuscript and provide thorough description of results/discussion. I do not wish to sound over-critical; the results are nice and believe the dissemination will be broader after additional refinement.

Comments/Questions:

- On Page 1, the authors state 'Antiskyrmions with an additional iteration of a Néel- and a Bloch-type wall are called second order antiskyrmions'. Before this, there is no mention of first order antiskyrmions.
- On Page 1, the authors state 'The measured bubbles differentiate themselves from the theoretical ideal contrast by a deformed elliptical shape, due to asymmetric Neel domain walls.' Can the authors expand this? Up to this point, authors have made no mention of asymmetric Neel walls in bubble skyrmions. Does this mean the bubbles possess hybrid domain walls?
- For Figure 1, How were the LTEM contrast images simulated? Did the authors assume a 2-dimensional or 3-dimensional magnetic spin textures? The latter affects the numerically expected contrast profile.
- On Page 2, the authors state 'In the LTEM images, we identify four distinct magnetic phases before saturation which also overlap.' Inspecting the images, I observed two magnetic phases: (i) disordered stripes and (ii) stripes coexisting with skyrmions ($S=-1, 0, 1$). Magnetic phases are commonly described in terms of long-range states.
- On Page 2, the authors state 'As the LTEM contrast does not reveal continuous Néel-type domain walls, it can be concluded that all stripes are separated by Bloch walls.' Normal-angle LTEM images do not necessarily show Néel-type contrast – they present the average-thickness in-plane magnetization which is primarily sensitive to Bloch-deflections – so making conclusion about the Néel character is not possible from solely inspecting under/over-focused LTEM images.

- For Figure 2 – There is a cylindrical-like spin texture with equal distribution white-black contrast which is not addressed. Inspecting the image, it does not appear to be a bubble, skyrmion nor antiskyrmion. Can the authors comment?
- For Figure 2 – The perpendicular loop shows a square-like hysteresis near zero-field. Is this a signature of the Fe/Gd/Ir specimen or an artifact of the magnetic specimen substrate side-walls? If the latter, the magnetization needs to be adjusted.
- On Page 3, the following statement can be misleading: 'It is also evident that pronounced opening of the oop hysteresis loop matches well with the existence of various spin objects'. The correlation is not necessarily vice-versa. The hysteresis near magnetic saturation describes the magnetic transformation of stripes to cylindrical spin textures [Refs. A. P. Malozemoff, et al. *Magnetic domain walls in bubble materials*, Academic Press; A. Hubert, R. Schafer, *Magnetic Domains*, etc.], but it does not imply a specific chiral spin texture are present.
- On Page 3, the authors state: 'As already mentioned, the domain morphology and nucleation centers of the different spin objects seem to be randomly distributed for every new field sweep starting from zero field after saturation'. What do you mean the nucleation centers? Do you mean the antiskyrmion do not form due at/near microstructural/magnetic-defects?
- On Page 4, the authors state 'For five of the six samples, it was possible to stabilize antiskyrmions even at room temperature.' I believe this needs to be further expanded. Up to this point, only [Fe/Ir/Gd]₂/ [Fe/Gd]₇₆/ [Fe/Ir/Gd]₂ has only been presented and no prior information has been detailed in the main manuscript regarding these other specimens. Less than one paragraph is dedicated to results obtained from the other 5 specimens which exhibit variations in the spin texture morphologies and magnetic properties.
- On Page 4, the authors state 'magnetic stripe domains with different types of chirality exist in the relaxed state at zero field'. Inspecting the domains states in Fig. 4(a), I was unable to observe any long-range Bloch-line chirality; instead, there are countless Bloch-points along the stripe domain wall.
- For Figs. 4, 5, 14, and 15, the authors should provide further information about the z-position shown for their micromagnetic simulation images. Is this the top-surface, center of the film? Also, given the relatively thick specimen, do the magnetic spin textures exhibit a hybrid domain wall? What is the expected 3-dimensional characteristics for the domain states?
- On Page 4, the authors state: 'Because of that, we can not rule out their existence at other temperature and field but conclude that they represent a metastable state in our system'. All field-stabilized spin textures (i.e., spin textures that require a constant applied magnetic field) are metastable states.
- Figure 3 and Supp. Fig. 11 show temperature- and field-dependent magnetic phase maps for different Fe/Gd-based multilayers. How did the authors decide to shade specific regions (e.g., blue, gray, purple) of the phase map where no experimental data was collected?
- On Page 11, the authors state 'Potential skyrmion phases can be identified by openings in the oop loops close to saturation.' As described by many clasworks on bubble materials [A. P. Malozemoff, et al. *Magnetic domain walls in bubble materials*, Academic Press; A. Hubert, R. Schafer, *Magnetic Domains*, etc.], the hysteresis commonly describes the phase transformation from stripes-to-cylindrical spin textures, but it does not imply a specific chiral spin texture is present.
- For most figures in the main text, the figure legend description seems insufficient. Suggest the

authors expand the description so readers know what is being performed/described.

- Experiments showed second-order antiskyrmions can be stabilized in select Fe/Gd multilayer compositions, but the authors never explicitly addressed differences in material properties that enabled first/second order type antiskyrmions to form. Can second order antiskyrmions form in select reproducible material conditions?

Reviewer #2:

Remarks to the Author:

The authors have experimentally shown the emergence of several spin textures in Fe/Gd-based multilayers with Ir insertion layers. In addition to non-topological type-II bubbles, they observe topological skyrmions, antiskyrmions and second-order antiskyrmions. They point out that the latter have previously never been experimentally observed, and antiskyrmions have yet to be experimentally stabilized in systems other than materials with D2D symmetry. They provide defocused Lorentz transmission electron microscopy (LTEM) micrographs of the spin textures as well as drawings of the theoretically predicted magnetizations and simulations of the expected LTEM contrast. They analyze the experimentally measured LTEM contrast of the initial lowest magnetization and recognize that certain conjunctions of the domain walls in this system result in Bloch lines which the authors observe are necessary for the formation of antiskyrmions. They go on to measure both the B-T and Ku-Ms phase diagrams of these materials.

The experimental results are novel and of interest to the general community, especially with the current burst of antiskyrmion research. In particular, they offer a new material class to explore antiskyrmions in. I believe the work deserves to be published, although I have several comments and questions that I believe need to be addressed before publication. I put the smaller comments and suggestions at the end.

1. Pg. 1 l. 53: It would be good to define second order antiskyrmions topological charge somewhere in the manuscript, as well as cite previous works predicting them, such as Desplat et al., PRB (2019), for example.
2. Pg. 1 l. 72: It isn't clear what the authors mean by 'deformed' in this context. Is it geometry? Are the authors attributing value to one geometrical shape over another? If so, can they provide evidence or reasoning that such a geometric bias is desirable?
3. Pg. 2 ls. 126-129: It isn't clear what the authors mean by asymmetric Néel domain walls, since the spin texture in Fig. 1d III looks Bloch-type, and there would not be any contrast if the domain walls were Néel-type, as the authors argue themselves on page 2, lines 135-138. Additionally, it isn't clear the measured contrast in Fig. 1d I is elliptical. Isn't it possible that the weak contrast at the edge of the spin texture is lost in the noise? It might be useful to include at least Poisson noise within the simulated images to more clearly compare the expected contrast with experiment. Also, the appearance of a "donut" hole contrast in Fig. 1b-c II suggests that the simulated defocus is different from experiment. This difference may cause confusion and is unnecessary. Can the authors change the simulation parameters to match the experimental defocus? Finally, I checked and unless I missed it somewhere, the authors don't seem to report their methods for simulating the LTEM contrast. Was a 2D approximation made or was 3D multislice employed? What were the input transmission functions for each spin texture?
4. Pg. 2 l. 158-164: This analysis is fine, but it may be worthwhile to perform a focal series and calculate the phase and magnetic inductance maps using the transport-of-intensity equation to guide the reader in following the conclusion.
5. Pg. 2 l. 162: What is the origin of the reduced LTEM contrast? The authors' analysis of the contrast reduction isn't sufficient to explain it. For example, they are right that antiparallel in-plane moments at two opposing domain walls would result in increased contrast, but only for one side of defocus. For the opposite defocus, the electrons should form weaker bright contrast away from the original bright line, resulting in the "large dark stripes" the authors see.

Additionally, if the author's interpretation that the weaker contrast stripes are due to parallel-oriented domain wall in-plane magnetizations, these magnetizations should meet somewhere, forming a Bloch line. Perhaps the formation of these Bloch lines could facilitate the formation of antiskyrmions? The authors should either provide a magnetic inductance map pointing out these features or provide higher magnification LTEM images at several defocus values including close to focus so that the reader may gain a better understanding of the magnetization within the domain walls.

6. Pg. 2 l. 166: This statement isn't clear. Are the authors attempting to describe the magnetization direction within the domain walls? A magnetic inductance map would clear up this issue.

7. Pg. 2 l. 180-183: Why do the authors claim that "it is evident?" It's a nice idea that the skyrmions form from the shrinking of the same chiral stripe domain, but it isn't entirely convincing from just the images presented in Fig. 2b. Can the authors indicate which stripes shrink into which skyrmions? I think I can point this out for at least one bubble on my own, but the skyrmions are not so clear.

8. Pg. 3 Fig. 1f III: Could the authors please increase the arrow shaft length and number of arrows in order to better see the magnetizations within the second-order antiskyrmion? As this spin texture is experimentally new, it would be good to more clearly illustrate it.

9. Pg. 5 l. 269-276: The contrast that the authors describe in lines 263-264 is a magnetic spin texture called a Bloch line. It has a long history, and four of them exist within an antiskyrmion. I recommend that the authors include Bloch lines in their paper, since they conclude that "the antiskyrmion always nucleated at the end of a stripe without chirality." I think this is a clear suggestion that Bloch lines are necessary to facilitate the generation of an antiskyrmion from a "stripe," and a nice result! It is clear that the Bloch line exists in the simulated magnetizations in Fig. 5e-h as well, as the authors point out.

10. Pg. 5 l. 293-294: The authors note that bulk DMI may exist, but did they consider it in simulations?

Smaller suggestions.

1. The results section on pg. 2 might be more reader-friendly to break into a couple of paragraphs.

2. Lines 162 and 172 seem to contradict each other regarding the periodicity.

3. Pg 4 Fig 4: This may not be necessary, but it would be clearer, and perhaps more aesthetically pleasing to increase the number of vector arrows drawn in the domain walls of Fig. 4.

4. Pg. 6 l. 380,402: These references are repeated.

5. Pg. 8 l. 580: Was all imaging performed at this defocus only? It would be a good idea to perform some lower defocus values as well to try to image some of the finer magnetic structure present within the domain walls.

**Response to Referees Letter Nature Communications manuscript NCOMMS-
20-43292-T**

**Dipolar-stabilized first and second-order antiskyrmions in ferrimagnetic
multilayers**

Dear Reviewers,

First, we would like to thank you for your valuable feedback. In the following, please find our reply addressing the comments/questions:

Reviewer #1

Revisions to the overall syntax are suggested to better communicate the presented results, as well as expand the subject matter in the main manuscript. Since Nature Communications facilitates publishing works with up to 5000 words (not including abstract, methods, references and figure legends), I suggest the authors bring results from the supplementary into the main manuscript and provide thorough description of results/discussion. I do not wish to sound over-critical; the results are nice and believe the dissemination will be broader after additional refinement.

By answering and addressing the comments and suggestions of the reviewers, we increase the word count from the original ~ 3500 words to ~ 4000 words in the new manuscript. For most figures the length of the figure texts was also significantly increased. We expanded the micromagnetic simulations of the domain morphology in Fig. 4 by adding data from the supplementary information. The new larger scale images emphasize the agreement between experimental and theoretical data even more and further help to understand the domain morphology. We kept our focus on one sample of the series but re-wrote/structured some paragraphs to emphasize that the other samples only reinforce the results of the shown sample and that the others show no new features.

Comments/Questions:

- (1) On Page 1, the authors state 'Antiskyrmions with an additional iteration of a Néel- and a Bloch-type wall are called second order antiskyrmions'. Before this, there is no mention of first order antiskyrmions.**

We addressed this issue by specifying the topological properties of first- and second-order antiskyrmions more clearly. We also added a reference to the theoretical prediction of second-order antiskyrmions:

*“Another novel magnetic quasi-particle that gained a lot of interest are magnetic antiskyrmions. They carry a negative vorticity (m) instead of the positive m of skyrmions. Thus, the topological charge (N_{sk}) of a skyrmion and antiskyrmion with the same polarity (p) is opposite to each other ($N_{sk}=p*m$). These first-order antiskyrmions exhibit a two-fold symmetry and consist of alternating Néel- Bloch-type domain walls that confine inside the out-of-plane magnetic moments from the surrounding antiparallel moments.” (line 41-49)*

...

“Antiskyrmions with an additional iteration of a Néel- and a Bloch-type wall are called second-order antiskyrmions. They have the vorticity $m=-2$ and show a three-fold symmetry. While they have been theoretically predicted [31], they have not been observed experimentally yet.” (line 56-60)

- (2) On Page 2, the authors state ‘The measured bubbles differentiate themselves from the theoretical ideal contrast by a deformed elliptical shape, due to asymmetric Neel domain walls.’ Can the authors expand this? Up to this point, authors have made no mention of asymmetric Neel walls in bubble skyrmions. Does this mean the bubbles possess hybrid domain walls?**

We clarified this statement with the following changes:

“Type-2 bubbles differentiate themselves from skyrmions by domain walls consisting of both Néel and Bloch type pointing roughly in one ip direction. Because of this, they also exhibit no rotational symmetry. The elliptical shape in Fig.1 d I and II arises from the one-fold symmetry of the spin configuration displayed in Fig. 1 d III.” (line 138-143)

- (3) For Figure 1, How were the LTEM contrast images simulated? Did the authors assume a 2-dimensional or 3-dimensional magnetic spin textures? The latter affects the numerically expected contrast profile.**

We addressed this issue by expanding the following paragraph and moving it from the method section to the results:

“The theoretical contrast of the spin textures in Fig. 1 b-f II was simulated by calculating the cross product of the two-dimensional in-plane (ip) components of the spin objects (Fig. 1 b-f pane II) and the incident electron beam multiplied by a deflection factor. The resulting smoothed histograms are plotted in Fig. 1 b-f (panel II). “ (line 122-127)

Additionally, we made the used Python code available on GitHub:

“Code availability

The Python code used to simulate the spin object contrasts is available under [5].” (line 704-708)

- (4) On Page 2, the authors state ‘In the LTEM images, we identify four distinct magnetic phases before saturation which also overlap.’ Inspecting the images, I observed two magnetic phases: (i) disordered stripes and (ii) stripes coexisting with skyrmions ($S=-1, 0, 1$). Magnetic phases are commonly described in terms of long-range states.**

It is true that “magnetic phase” was the wrong term to describe the different ranges of stability of our spin objects. The manuscript was changed accordingly:

“In the LTEM images, we identify different ranges of stability of the different spin objects.” (line 173-174)

- (5) On Page 2, the authors state ‘As the LTEM contrast does not reveal continuous Néel-type domain walls, it can be concluded that all stripes are separated by Bloch walls.’ Normal-angle LTEM images do not necessarily show Néel-type contrast – they present the average-thickness in-plane magnetization which is primarily sensitive to Bloch-deflections – so making conclusion about the Néel character is not possible from solely inspecting under/over-focused LTEM images.**

We also tilted our sample as mentioned in the end of this paragraph to exclude possible pure Néel textures:

“Jiang et al. [42] showed that Néel skyrmions exhibit no LTEM contrast when the film is normal to the electron beam. Considering this, we tilted our film samples by up to 30 degrees revealing no indication of the existence of pure Néel-type spin objects.” (line 149-153)

But we still agree that this sentence is misleading at this point and deleted it.

- (6) For Figure 2 – There is a cylindrical-like spin texture with equal distribution white-black contrast which is not addressed. Inspecting the image, it does not appear to be a bubble, skyrmion nor antiskyrmion. Can the authors’ comment?**

Thanks for noticing that. This object is a CCW skyrmion (white dot) pinned to a micro-structural defect (smaller black dot). The smaller black dot is visible in all six panels of Fig. 2 and shows no changes under magnetic fields. The text addresses this feature now:

“All six panels also show micro-structural defects as black dots which are a slightly smaller than the skyrmions and unaffected by the magnetic field. In Fig. 2 b II-V one of the counterclockwise skyrmion is pinned to one of these defects.” (line 211-215)

- (7) For Figure 2 – The perpendicular loop shows a square-like hysteresis near zero-field. Is this a signature of the Fe/Gd/Ir specimen or an artifact of the magnetic specimen substrate side-walls? If the latter, the magnetization needs to be adjusted.**

This feature is also visible in MOKE measurements, that this is not an artifact of the substrate or sample geometry. In a recent publication (Mandru, A.-O. et al., Journal of Vacuum Science & Technology A 38, 023409 (2020)) we discussed in more detail substrate side wall effects.

- (8) On Page 3, the following statement can be misleading: ‘It is also evident that pronounced opening of the oop hysteresis loop matches well with the existence of various spin objects.’ The correlation is not necessarily vice-versa. The hysteresis near magnetic saturation describes the magnetic transformation of stripes to cylindrical spin textures [Refs. A. P. Malozemoff, et al. Magnetic domain walls in bubble materials, Academic Press; A. Hubert, R. Schafer, Magnetic Domains, etc.], but it does not imply a specific chiral spin texture are present.**

We address this issue by the following additional sentence:

“At larger fields, the hysteresis loop opens because of irreversible processes, like the formation and annihilation of cylindrical spin textures [43].” (line 170-173)

- (9) On Page 3, the authors state: ‘As already mentioned, the domain morphology and nucleation centers of the different spin objects seem to be randomly distributed for every new field sweep starting from zero field after saturation’. What do you mean the nucleation centers? Do you mean the antiskyrmion do not form due at/near microstructural/magnetic-defects?**

Yes, we wanted to express that the different spin objects do not seem to nucleate at certain spots because of structural inhomogeneities. To make this clearer, the sentence was modified in the following way:

“As already mentioned, the domain morphology and position of the different spin objects seem to be randomly distributed for every new field sweep starting from zero field after saturation. This leads us to the conclusion that neither structural nor magnetic inhomogeneities are the reason for the variety of different spin objects.” (line 226-231)

- (10) On Page 4, the authors state ‘For five of the six samples, it was possible to stabilize antiskyrmions even at room temperature.’ I believe this needs to be further expanded. Up to this point, only [Fe/Ir/Gd]_{x2}/ [Fe/Gd]_{x76}/ [Fe/Ir/Gd]_{x2} has only been presented and no prior information has been detailed in the main manuscript regarding these other specimens. Less than one paragraph is dedicated to results obtained from the other 5 specimens which exhibit variations in the spin texture morphologies and magnetic properties.**

The overall information we gathered is basically the same for all samples: First order antiskyrmions only coexist with skyrmions, second-order antiskyrmions are very rare. To keep the manuscript focused, we chose to pick one of the specimen, but we still wanted to acknowledge that our findings can be observed as well in the other samples which show slightly different values in K_u and M_s which were actually used to create the phase map in Fig. 6 containing all samples.

But we also think that this paragraph could benefit from a clearer separation to the supplementary information and added some additional information of the other specimen in the following way:

“We repeated this procedure for different temperatures between 100 and 300 K. The resulting magnetic phase diagram of $[Fe/Ir/Gd]_2 / [Fe/Gd]_{76} / [Fe/Ir/Gd]_2$ is displayed in Fig.3. It was constructed using the data from the LTEM measurements marked by the black dots. The area in between the experimental data was filled under the assumption that stability transitions happen exactly in the middle of two data points. As already mentioned, the domain morphology and position of the different spin objects seem to be randomly distributed for every new field sweep starting from zero field after saturation. This leads us to the conclusion that structural or magnetic inhomogeneities cannot be the reason for the variety of different spin objects. At the same time, the range of stability of the different spin objects stayed the same for every field sweep. Further, we did not observe a pure antiskyrmion phase. They always coexisted with Bloch skyrmions and sometimes with type-2 bubbles. It was also evident that the stability range of antiskyrmions is smaller than the one of skyrmions for both temperature and field. Second-order antiskyrmions were only observable at 260 K but stayed stable for larger magnetic fields in comparison to first-order antiskyrmions.

We also measured and created phase diagrams of the five other samples of our series ($N_{Ir}=0, 5, 10, 20, 40$). The other phase diagrams (SI-Fig.11), hysteresis loops (SI-Fig.9), and additional LTEM images (SI-Fig.12) are available in the supplementary information. Generally, more Ir insertion layers decrease both magnetization and uniaxial magnetic anisotropy (SI-Fig.10). Because of a too low magnetization, the sample with $N_{Ir}=40$ was not possible to image by LTEM. The other five samples showed antiskyrmions at room temperature. While samples with $N_{Ir}=0,2,10$, and 20 only show antiskyrmions at larger temperatures, the sample with $N_{Ir}=5$ exhibits antiskyrmions at all measured temperatures. First-order antiskyrmions were observed at a wide range of temperatures in every sample of our series besides $N_{Ir}=40$, however, second-order antiskyrmions were only observed at 260 K in $N_{Ir}=2$ and at 300 K in $N_{Ir}=20$. In all cases, they were quite rare in comparison to the other spin objects. Because of that, we cannot rule out their existence at other temperatures and fields but conclude that their nucleation process is less likely in our samples in comparison to first-order (anti-)skyrmions. “ (line 219-261)

- (11) On Page 4, the authors state ‘magnetic stripe domains with different types of chirality exist in the relaxed state at zero field’. Inspecting the domains states in Fig. 4(a), I was unable to observe any long-range Bloch-line chirality; instead, there are countless Bloch-points along the stripe domain wall.**

We expanded Fig. 4 by adding data from the supplementary information. The new larger scale images clarify the long-range domain morphology.

We also added a more detailed description of the states, also mentioning the Bloch-points:

“Fig.4 a and e display the relaxed magnetization state at zero field showing stripes with different chiralities bristled with Bloch points...” (line 282-284)

- (12) For Figs. 4, 5, 14, and 15, the authors should provide further information about the z-position shown for their micromagnetic simulation images. Is this the top-surface, center of the film? Also, given the relatively thick specimen, do the magnetic spin textures exhibit a hybrid domain wall? What is the expected 3-dimensional characteristics for the domain states?

Our micromagnetic simulations show the top surface, but we could not observe any significant difference to the center or bottom of the films. We are aware that other similar systems showed a Néel capping of their Bloch skyrmions in their simulations [Montoya, S. A. et al. Phys. Rev. B 95, 2024415 (2017)]. We attribute this difference to the lower magnetization of our samples. The following sentence was added to the manuscript:

“The simulations did not show significant differences in their spin configuration along the film thickness.” (line 277-278)

- (13) On Page 4, the authors state: ‘Because of that, we cannot rule out their existence at other temperature and field but conclude that they represent a metastable state in our system’. All field-stabilized spin textures (i.e., spin textures that require a constant applied magnetic field) are metastable states.

This is true, we changed the sentence accordingly to be more specific:

“Because of that, we cannot rule out their existence at other temperatures and fields but conclude that their nucleation process is less likely in our samples in comparison to first-order antiskyrmions.” (line 258-261)

- (14) Figure 3 and Supp. Fig. 11 show temperature- and field-dependent magnetic phase maps for different Fe/Gd-based multilayers. How did the authors decide to shade specific regions (e.g., blue, gray, purple) of the phase map where no experimental data was collected?**

We added the following statement:

“The diagram was constructed using the data from the LTEM measurements marked by the black dots. The area in between the experimental data was filled under the assumption that phase transitions happen in the middle of the measured data points.” (line 222-226)

- (15) On Page 11, the authors state ‘Potential skyrmion phases can be identified by openings in the oop loops close to saturation.’ As described by many clasworks on bubble materials [A. P. Malozemoff, et al. Magnetic domain walls in bubble materials, Academic Press; A. Hubert, R. Schafer, Magnetic Domains, etc.], the hysteresis commonly describes the phase transformation from stripes-to-cylindrical spin textures, but it does not imply a specific chiral spin texture is present.**

We agree and softened this statement:

“The formation of spin textures is sometimes revealed by the presence of an opening in the oop loop close to saturation” (line 777-778)

- (16) For most figures in the main text, the figure legend description seems insufficient. Suggest the authors expand the description so readers know what is being performed/described.**

We added more detailed figure legends for all figures.

- (17) Experiments showed second-order antiskyrmions can be stabilized in select Fe/Gd multilayer compositions, but the authors never explicitly addressed differences in material properties that enabled first/second order type antiskyrmions to form. Can second order antiskyrmions form in select reproducible material conditions?**

In this manuscript we tried to focus on the necessary magnetic parameters to stabilize first-order antiskyrmion, as displayed in Fig. 6, but we also wanted to mention the existence of second-order antiskyrmions in this system. Because of the rarity of second-order antiskyrmions we could not gather enough experimental data to give a profound recipe. This will be a subject of further studies. The following addition describes the nature of our second-order antiskyrmion observation clearer:

“Because of that, we cannot rule out their existence at other temperatures and fields but conclude that their nucleation process is less likely in our samples in comparison to first-order antiskyrmions.” (line 258-261)

Reviewer #2

- 1. Pg. 1 l. 53: It would be good to define second order antiskyrmions topological charge somewhere in the manuscript, as well as cite previous works predicting them, such as Desplat et al., PRB (2019), for example.**

We addressed this issue by specifying the topological properties of first- and second-order antiskyrmions more clearly and cited the suggested previous work:

*“Another novel magnetic quasi-particle that gained a lot of interest are magnetic antiskyrmions. They carry a negative vorticity (m) instead of the positive m of skyrmions. Thus, the topological charge (N_{sk}) of a skyrmion and antiskyrmion with the same polarity (p) is opposite to each other ($N_{sk}=p*m$). These first-order antiskyrmions exhibit a two-fold symmetry and consist of alternating Néel- and Bloch-type domain walls that confine inside the out-of-plane magnetic moments from the surrounding antiparallel moments.” (line 41-49)*

“Antiskyrmions with an additional iteration of a Néel- and a Bloch-type wall are called second-order antiskyrmions. They have the vorticity $m=-2$ and show a three-fold symmetry. While they have been theoretically predicted [31], they have not been observed experimentally yet.” (line 56-60)

- 2. Pg. 1 l. 72: It isn't clear what the authors mean by 'deformed' in this context. Is it geometry? Are the authors attributing value to one geometrical shape over another? If so, can they provide evidence or reasoning that such a geometric bias is desirable?**

Yes, both cited works showed geometrical deformed squared antiskyrmions and elliptical skyrmions. The elliptical skyrmions are not rotational symmetric and oriented along crystal directions, as well as antiskyrmions. This geometrical confinement could be an issue for future experiments of antiskyrmion-skyrmion interaction or current driven motion. We changed the sentence to increase clarity:

“In these crystalline bulk systems, the spin objects exist only in a geometrical deformed state and in specific planes of the crystal.” (line 77-79)

- 3. Pg. 2 ls. 126-129: It isn't clear what the authors mean by asymmetric Néel domain walls, since the spin texture in Fig. 1d III looks Bloch-type, and there would not be any contrast if the domain walls were Néel-type, as the authors argue themselves on page 2, lines 135-138. Additionally, it isn't clear the measured contrast in Fig. 1d I is elliptical. Isn't it possible that the weak contrast**

at the edge of the spin texture is lost in the noise? It might be useful to include at least Poisson noise within the simulated images to more clearly compare the expected contrast with experiment. Also, the appearance of a "donut" hole contrast in Fig. 1b-c II suggests that the simulated defocus is different from experiment. This difference may cause confusion and is unnecessary. Can the authors change the simulation parameters to match the experimental defocus?

Finally, I checked and unless I missed it somewhere, the authors don't seem to report their methods for simulating the LTEM contrast. Was a 2D approximation made or was 3D multislice employed? What were the input transmission functions for each spin texture?

- Type-2 bubbles are no longer necessary round because of their missing rotational symmetry. We made this statement clearer with the following changes:

"Type-2 bubbles differentiate themselves from skyrmions by domain walls consisting of both Néel and Bloch type pointing roughly in one ip direction. Because of this, they also exhibit no rotational symmetry. The elliptical shape in Fig.1 d I and II arises from the one-fold symmetry of the spin configuration displayed in Fig. 1 d III." (line 138-143)

- In LTEM continuous Néel domain walls are not visible, this is not the case in a type-2 bubble.
- To improve clarity how we created our simulation images, we expanded the following section and moved it from the method section to the results:

"The theoretical contrast of the spin textures in Fig. 1 b-f II was simulated by calculating the cross product of the two-dimensional in-plane (ip) components of the spin objects (Fig. 1 b-f pane II) and the incident electron beam multiplied by a deflection factor. The resulting smoothed histograms are plotted in Fig. 1 b-f (panel II). " (line 122-127)

In fact, we were quite happy with the simulation results and how close we got to the experimental data with this simple approach. For all images the same deflection factor (simulated defocus) was used. Additionally, we made the programmed Python code available on GitHub: <https://github.com/Julian-Hi/LTEM-contrast>

4. **Pg. 2 I. 158-164: This analysis is fine, but it may be worthwhile to perform a focal series and calculate the phase and magnetic inductance maps using the transport-of-intensity equation to guide the reader in following the conclusion.**

We fully agree with this. Unfortunately, the used FEI Titan microscope did not have the option to capture both images in under and over-focus. Our new JEOL system has this option, but we do not have the software ready to create magnetic inductance maps from it. But we look forward to doing this for future work. Also, DPC images could be very interesting and we are working on that.

5. **Pg. 2 I. 162: What is the origin of the reduced LTEM contrast? The authors' analysis of the contrast reduction isn't sufficient to explain it. For example, they are right that antiparallel in-plane moments at two opposing domain walls would result in increased contrast, but only for one side of defocus. for the opposite defocus, the electrons should form weaker bright contrast away from the original bright line, resulting in the "large dark stripes" the authors see. Additionally, if the author's interpretation that the weaker contrast stripes are due to parallel-oriented domain wall in-plane magnetizations, these magnetizations should meet somewhere,**

forming a Bloch line. Perhaps the formation of these Bloch lines could facilitate the formation of antiskyrmions? The authors should either provide a magnetic inductance map pointing out these features or provide higher magnification LTEM images at several defocus values including close to focus so that the reader may gain a better understanding of the magnetization within the domain walls.

As the reviewer mentions, stripes with chiralities act to the electrons of the TEM in the chosen defocus value like a focus or defocus lens. The electrons are deflected by the ip components of a stripe to the center (bright line) or away from the center. The resulting contrast stems from both ip domain walls surrounding the oop components. One object roughly the size of the periodicity of the magnetic stripe is visible. In contrast a stripe without a chirality has ip components pointing in the same direction. This leads to a dark and bright line for every domain wall. Stipes with half the periodicity of the magnetic stripes are observable. Because of that, we observe roughly half the contrast of the stripes with chirality. These properties are already well described in the literature (Ref. 16,17). But we additionally cited a very recent work of Garlow et al. This work is purely focused on the phenomena of stripes with different chiralities and their interaction investigated by LTEM.

"This state of stripe domains with mixed chiralities is describe in more detail in the work of Garlow et al. [43] and was also found as the ground state in similar thin-film systems [16, 17]." (line 186-189)

We took a closer look at the ends of stripes without chirality and Bloch lines in the section where we discuss the nucleation process of our antiskyrmions. This focuses more on the aspect of Bloch lines as nucleation centers for antiskyrmions. We provide here in Fig. 5 LTEM images with higher magnification. Images captured closer to the defocus suffer immensely from small contrast because of the small magnetic moments our samples show. We think the shown images are the best compromise between contrast and detail in domain wall morphology. The mentioned Bloch lines are clearly visible in Fig. 5 and we think the very comparable magnetic simulation images help compensate the lack of magnetic inductance maps.

6. Pg. 2 l. 166: This statement isn't clear. Are the authors attempting to describe the magnetization direction within the domain walls? A magnetic inductance map would clear up this issue.

We agree that magnetic inductance map would help here, but stripes with their different chirality are not the focus of this work. The cited work [17,18,43] describes well with magnetic inductance maps, how different chiralities form the characteristic broader and thinner stripes in LTEM.

7. Pg. 2 l. 180-183: Why do the authors claim that "it is evident?" It's a nice idea that the skyrmions form from the shrinking of the same chiral stripe domain, but it isn't entirely convincing from just the images presented in Fig. 2b. Can the authors indicate which stripes shrink into which skyrmions? I think I can point this out for at least one bubble on my own, but the skyrmions are not so clear.

We agree that more images with smaller field steps would be more convincing. Due to our focus on antiskyrmions and their nucleation process, we did not go into more detail here. In the LTEM

images in the supplementary information better examples for this process can be found in Fig. 12 a-d) and i-l), as well as in the micromagnetic simulations in Fig. 14. It is important to note that like the nucleation process of antiskyrmions above a temperature of 250 K spin objects do not necessarily need a certain domain morphology to nucleate. We also observe the switching of the chiralities of stripes while applying a field, if the stripes have crossings. All these details lead us to weaken our statement:

"The different chiralities of the stripes can determine the type of spin objects that can form under oop magnetic field." (line 189-191)

"We observe that they preferably arise from collapsed stripe domains with the same chirality. Equal numbers of counterclockwise (white) and clockwise (black) Bloch skyrmions appear when averaged over the whole sample." (line 194-198)

8. **Pg. 3 Fig. 1f III:** Could the authors please increase the arrow shaft length and number of arrows in order to better see the magnetizations within the second-order antiskyrmion? As this spin texture is experimentally new, it would be good to more clearly illustrate it.

We increased the arrow shaft length which improved the clarity of the second-order antiskyrmions greatly:

9. **Pg. 5 I. 269-276:** The contrast that the authors describe in lines 263-264 is a magnetic spin texture called a Bloch line. It has a long history, and four of them exist within an antiskyrmion. I recommend that the authors include Bloch lines in their paper, since they conclude that "the antiskyrmion always nucleated at the end of a stripe without chirality." I think this is a clear suggestion that Bloch lines are necessary to facilitate the generation of an antiskyrmion from a "stripe," and a nice result! It is clear that the Bloch line exists in the simulated magnetizations in Fig. 5e-h as well, as the authors point out.

We expanded on that with a more detailed Fig. 4. Bloch lines and points are now mentioned multiple times in the manuscript, especially in the section describing the nucleation of antiskyrmions:

"Fig.4 a and e display the relaxed magnetization state at zero field showing stripes with different chiralities bristled with Bloch points." (line 282-284)

"Three stripes without a chirality meet and due to the resulting in-plane components, one stable Bloch point emerges resulting in a contrast very similar to an antiskyrmion." (line 310-312)

10. Pg. 5 l. 293-294: The authors note that bulk DMI may exist, but did they consider it in simulations?

In the shown simulations in the main manuscript we did not include DMI, but we mention that the measured DMI values of our films only have a minor impact on the results. Additional simulation results with DMI and depending on DMI are included in the supplementary information.

Smaller suggestions:

i. The results section on pg. 2 might be more reader-friendly to break into a couple of paragraphs.

We addressed this by adding additional paragraphs.

ii. Lines 162 and 172 seem to contradict each other regarding the periodicity.

If a magnetic stripe has a chirality it appears with roughly its size as one dark or bright stripe in LTEM. If a stripe has no chirality it appears as two bright and dark narrower ones with half the size. The overall periodicity of the magnetic structure is still the one of the broader chiral stripes.

iii. Pg 4 Fig 4: This may not be necessary, but it would be clearer, and perhaps more aesthetically pleasing to increase the number of vector arrows drawn in the domain walls of Fig. 4.

We tried this. With a denser array of vectors, the readability worsens.

iv. Pg. 6 l. 380,402: These references are repeated.

Thanks, this error was fixed.

v. Pg. 8 l. 580: Was all imaging performed at this defocus only? It would be a good idea to perform some lower defocus values as well to try to image some of the finer magnetic structure present within the domain walls.

Yes, all measurements were done at this larger defocus. It was the best compromise between contrast and resolution. We agree that for future work it would be interesting to go into more detail of the finer magnetic structure together with magnetic inductance maps. Also, DPC imaging will be very interesting on these structures.

Additional changes

- The statements *“Skyrmions and antiskyrmions are topologically protected spin structures with opposite topological charge.”* and *“They carry the integer topological charge of $N_{sk} = -1$ instead of the $N_{sk} = +1$ of skyrmions.”* were generalized. Skyrmions and antiskyrmions only have an opposite topological charge if they have the same polarity. The statements were changed in the following way:

“Skyrmions and antiskyrmions are topologically protected spin structures with opposite vorticities.” (line 2)

*“Another novel magnetic quasi-particle that gained a lot of interest are magnetic antiskyrmions. They carry a negative vorticity (m) instead of the positive m of skyrmions. Thus, the topological charge (N_{sk}) of a skyrmion and antiskyrmion with the same polarity (p) is opposite to each other ($N_{sk}=p*m$).” (line 41-45)*

- A data availability statement was added to the end of the method section.
- A code availability statement was added to the end of the method section. The Python code we programmed for the theoretical contrast of our spin structures is now available on GitHub: <https://github.com/Julian-Hi/LTEM-contrast>

Reviewers' Comments:

Reviewer #1:

None

Reviewer #2:

Remarks to the Author:

The authors response has improved the work immensely. However, I have more questions and comments that should be addressed. While I believe the results are novel and deserve to be published, I have to agree with the first reviewer that this manuscript is not refined and as a result reads in some places as less than rigorous. I believe this can and should be improved before publication in Nature Communications.

Just as a side note, it seems as the first reviewer used the word "long-range Bloch-line to describe the domain wall length. I want to be clear here that this is distinct from the vertical Bloch lines I mention here, which exist as a line throughout the thickness of the magnet along the domain wall. I describe Bloch points, in contrast, as a magnetic point 'defect' along a line of spin moments. The latter would be near impossible to see in LTEM under the current large defocus value.

"We fully agree with this. Unfortunately, the used FEI Titan microscope did not have the option to capture both images in under and over-focus. Our new JEOL system has this option, but we do not have the software ready to create magnetic inductance maps from it. But we look forward to doing this for future work."

Perhaps this is ignorance, but I've never heard of a microscope that cannot perform over- and under-focus imaging, because it's just increasing or decreasing the current in the lens. There are also ways to calculate the magnetic induction map from a single TIE image as well (see, for example, Chess, J. et al., Ultramicroscopy 177, 78-83, (2017)), and I still think this paper could really use these induction maps to better support the authors' analysis. This all assumes that the authors can achieve contrast at lower defocus values, however, which may be difficult as they state their micrographs were obtained at 3.5 mm defocus. Assuming this is all true, and the authors are quite unlucky to be in such a predicament, I think some of the claims of the paper should now be dampened because the LTEM analysis is limited to one side of focus. For example, without a phase and/or magnetic induction map, it is dangerous to draw quantitative conclusions, including the authors' statement on page 2 (lns 184-6) that the "underlying periodicity of the magnetic stripes is about 250 nm and is the same for both types." I recommend removing the word "underlying", because it implies that we can estimate the magnetic domains periodicity quantitatively using large, single defocus LTEM, which is dubious because of the blending of contrast from small and large magnetic and electrostatic structures as well as the lack of a view from the other side of focus. If the authors could provide lower defocus images showing some of the finer structure of the magnetic domain wall locations, this statement may remain.

Additionally, if the authors use a lower defocus value, they could also raise or lower the sample height in order to achieve the other side of focus. For these same reasons, it is quite difficult to say with certainty that the "broader" stripes have a chirality while the "narrow" ones exhibit no chirality, as the authors claim on page 2 lns 182-4. Have the authors considered the possibility that there are two periodic stripe structures, both having chirality? As the authors point out, the nature of these stripe domains is not the focus of this work, so I recommend removing lines 178-186 "The bordering ... for both types," and just suffice to say that the while the technical limitations of the experimental apparatus limit the analysis of such magnetic domains, they seem to be similar to domains seen in previous works, described in detail in [44], etc...

"We expanded on that with a more detailed Fig. 4. Bloch lines and points are now mentioned multiple times in the manuscript, especially in the section describing the nucleation of antiskyrmions:"

Although the authors say they mention Bloch lines, a quick search reveals that they don't mention Bloch lines at all. Bloch points are, however, mentioned. I caution the authors on this, though, because Bloch points imply (and certainly have implied in the literature) a single Néel-type magnetic moment (or monopole) within a large Bloch-type domain wall, which would almost certainly be impossible to distinguish experimentally. However, Figure 5 shows clear contrast that there are Bloch lines, which must exist throughout the entire domain wall for the contrast to be on the order of the walls around it. Additionally, the authors mention that there are no significant differences throughout the thickness of their 62 nm thick magnetic simulations (lines 277-8), which implies that the Bloch points seen in the 2D slices in Figure 4 are actually Bloch lines as well. I recommend using this "line" language for clarity.

"As already mentioned, the domain morphology and position of the different spin objects seem to be randomly distributed for every new field sweep starting from zero field after saturation. This leads us to the conclusion that neither structural nor magnetic inhomogeneities are the reason for the variety of different spin object." (lines 226-31)

I'm not sure that there aren't any magnetic inhomogeneities. It's well known that magnets with a relatively large uniaxial anisotropy readily form type I (skyrmions) and type II bubbles. As the authors found with this new magnet, the reduction of uniaxial magnetic anisotropy leads to a "softening" of the magnet, and the introduction of many Bloch lines, which are seen both experimentally and via simulations. The data seems to suggest that these Bloch lines are the magnetic inhomogeneities needed to diversify the available spin objects.

Further questions that should be addressed:

1. On page 2 line 179 and again on 181, the authors claim that the in-plane components of the magnetization within the domain wall are "parallel to the stripes." This directly contradicts the direction of the in-plane magnetizations of the simulations in Figures 4 and 5. Why is this?
2. Why did the authors choose to employ a 2D LTEM model when their magnet is quite thick and would be better suited for multislice simulations, which would consider the full 3D structure of the spin textures? I recommend providing more of a detailed explanation of the contrast algorithm in the supporting materials, either as its own section or as part of the README file, which currently doesn't describe the details. I also recommend providing reasoning for why such a simplified model is used, including the above mentioned 2D assumption.

**Second Response to Referees Letter Nature Communications manuscript
NCOMMS-20-43292A**

**Dipolar-stabilized first and second-order antiskyrmions in ferrimagnetic
multilayers**

Dear Reviewer,

Thank you again for your comments and feedback! We agree that the first revision immensely improved the manuscript, and we hope that our manuscript is ready for publication after this revision. In the following, please find our reply addressing the comments/questions:

Reviewer #2

Perhaps this is ignorance, but I've never heard of a microscope that cannot perform over- and under-focus imaging, because it's just increasing or decreasing the current in the lens. There are also ways to calculate the magnetic induction map from a single TIE image as well (see, for example, Chess, J. et al., *Ultramicroscopy* 177, 78-83, (2017)), and I still think this paper could really use these induction maps to better support the authors' analysis. This all assumes that the authors can achieve contrast at lower defocus values, however, which may be difficult as they state their micrographs were obtained at 3.5 mm defocus. Assuming this is all true, and the authors are quite unlucky to be in such a predicament,

We did not explain this clearly enough in the first response letter: The used FEI Titan system can, of course, perform over- and underfocus imaging, but it cannot do it with the large defocus values that were necessary in both directions (3.5 mm).

I think some of the claims of the paper should now be dampened because the LTEM analysis is limited to one side of focus. For example, without a phase and/or magnetic induction map, it is dangerous to draw quantitative conclusions, including the authors' statement on page 2 (Ins 184-6) that the "underlying periodicity of the magnetic stripes is about 250 nm and is the same for both types." I recommend removing the word "underlying", because it implies that we can estimate the magnetic domains periodicity quantitatively using large, single defocus LTEM, which is dubious because of the blending of contrast from small and large magnetic and electrostatic structures as well as the lack of a view from the other side of focus. If the authors could provide lower defocus images showing some of the finer structure of the magnetic domain wall locations, this statement may remain. Additionally, if the authors use a lower defocus value, they could also raise or lower the sample height in order to achieve the other side of focus. For these same reasons, it is quite difficult to say with certainty that the "broader" stripes have a chirality while the "narrow" ones exhibit no chirality, as the authors claim on page 2 Ins 182-4. Have the authors considered the possibility that there are two periodic stripe structures, both having chirality? As the authors point

out, the nature of these stripe domains is not the focus of this work, so I recommend removing lines 178-186 “The bordering ... for both types,” and just suffice to say that the while the technical limitations of the experimental apparatus limit the analysis of such magnetic domains, they seem to be similar to domains seen in previous works, described in detail in [44], etc...

We appreciate this suggestion as we cannot provide lower defocus LTEM imaging. However, to support our claim, we additionally measured the domain morphology on all samples by MFM which confirms only one underlying periodicity.

Fig. 12 Exemplary LTEM image (a) of [Fe/Gd]80 in comparison to the MFM image (b). Both images were captured in zero field at room temperature. Note that the images do not show the same region of the sample but share the same scale.

We added one exemplary MFM image (the figure above) to the supplementary information and this paragraph:

“To confirm the underlying magnetic periodicity of our samples, magnetic force microscopy images were captured. Figure 12 shows the LTEM image (Fig. 12 a) of [Fe/Gd]80 in comparison to an exemplary MFM image (Fig. 12 b). Both images were captured in zero field at room temperature. Note that the images do not show the same region of the sample but share the same scale. The size of the magnetic domains captured by MFM match roughly the size of the larger high-contrast black and white stripes in the defocused LTEM image.” (line 833-842)

Additionally, we modified our statement about the chirality of the stripes, as suggested by the reviewer:

“Starting from zero field, the film exhibits two different kinds of stripe domains up to 63 mT: broader dark and bright stripes and narrower stripes with less contrast and half the periodicity of the broader stripes (Fig. 2b I). While the technical limitations of the experimental apparatus limit the analysis of such magnetic domains, they seem to be similar to domains seen in previous works \cite{Montoya2017,Zhang2020}, described in detail in \cite{PhysRevB.102.214429}. The bordering in-plane components are aligned mostly parallel to the stripes. Antiparallel in-plane moments on the opposite sides of the stripe result in larger high-contrast stripes, while parallel aligned Bloch walls result in a narrower stripe pattern. In other words, the “broader” stripes exhibit domain walls with a chirality, while the “narrower” ones exhibit no chirality. However, the underlying periodicity of the magnetic

stripes is about 250 nm and the same for both types. The existence of only one underlying periodicity was also confirmed by additional magnetic force microscopy measurements (SI-Fig. 12). “ (line 174-192)

Although the authors say they mention Bloch lines, a quick search reveals that they don't mention Bloch lines at all. Bloch points are, however, mentioned. I caution the authors on this, though, because Bloch points imply (and certainly have implied in the literature) a single Néel-type magnetic moment (or monopole) within a large Bloch-type domain wall, which would almost certainly be impossible to distinguish experimentally. However, Figure 5 shows clear contrast that there are Bloch lines, which must exist throughout the entire domain wall for the contrast to be on the order of the walls around it. Additionally, the authors mention that there are no significant differences throughout the thickness of their 62 nm thick magnetic simulations (lines 277-8), which implies that the Bloch points seen in the 2D slices in Figure 4 are actually Bloch lines as well. I recommend using this “line” language for clarity.

We fully agree that the term Bloch line improves clarity. Thanks for noticing this important detail! We changed the manuscript both in the experimental as well as the simulation paragraphs.

I'm not sure that there aren't any magnetic inhomogeneities. It's well known that magnets with a relatively large uniaxial anisotropy readily form type I (skyrmions) and type II bubbles. As the authors found with this new magnet, the reduction of uniaxial magnetic anisotropy leads to a “softening” of the magnet, and the introduction of many Bloch lines, which are seen both experimentally and via simulations. The data seems to suggest that these Bloch lines are the magnetic inhomogeneities needed to diversify the available spin objects.

We certainly agree that our system exhibits magnetic inhomogeneities. We only wanted to convey with this statement that we could not find specific reoccurring nucleation points in the samples after multiple field cycles. Our micromagnetic simulations confirm this behavior and show that no inhomogeneities in the magnetic properties are necessary to nucleate our diversity of spin objects. But the reviewer is fully right that we cannot conclude experimentally just from this observation that magnetic inhomogeneities are not necessary. Thus, the sentence *“This leads us to the conclusion that neither structural nor magnetic inhomogeneities are the reason for the variety of different spin objects.”* was removed.

Further questions:

- 1. On page 2 line 179 and again on 181, the authors claim that the in-plane components of the magnetization within the domain wall are “parallel to the stripes.” This directly contradicts the direction of the in-plane magnetizations of the simulations in Figures 4 and 5. Why is this?**

It is true that the Néel-nature of the domain walls of the stripes in the initial state is more pronounced in the simulations in comparison to the measurements. This also leads to a more fractured domain morphology and more Bloch lines in the simulations. The observed LTEM patterns are still only possible for Bloch-type domain walls surrounding the stripes, but to address possible Néel components, we weakened our statement:

“The bordering in-plane components are aligned mostly parallel to the stripes.” (line 182-183)

- 2. Why did the authors choose to employ a 2D LTEM model when their magnet is quite thick and would be better suited for multislice simulations, which would consider the full 3D structure of the spin textures? I recommend providing more of a detailed explanation of the contrast algorithm in the supporting materials, either as its own section or as part of the README file, which currently doesn't describe the details. I also recommend providing reasoning for why such a simplified model is used, including the above mentioned 2D assumption.**

Our micromagnetic simulations showed no changes in the magnetic structure depending on the z-position in the film. We also think a 2D grid is a reasonable assumption since the film thickness is much smaller than the lateral dimensions. Additionally, our simple 2D model agrees very well with the observed contrasts in our experiments as well as previous observations and simulations of bubbles, skyrmions, and antiskyrmions (Nayak, A. et al. Magnetic antiskyrmions above room temperature in tetragonal Heusler materials. *Nature* 548, 561–566 (2017).). The contrast simulations were never meant to be a perfect representation of the experiments including all the details, but to show the theoretical contrast of the different possible spin objects.

Following the suggestion of the reviewer, we both expanded the README file and the method section describing our simple 2D model:

“The theoretical contrast of the spin textures in Fig. 1 b-f (panel II) was simulated in the following way: Parallel incoming electrons are modeled as vectors to impinge on the spin structure on a discrete 2D grid at an angle of 90°. Analytical functions of the ip component of the 2D spin objects are used to model the internal magnetic field as vectors. In the magnetic field of the structure, electrons are deflected by the Lorentz force into a direction perpendicular to their own velocity and the local magnetization direction of the spin structure. In the code, the cross product of the magnetic field vectors and the incoming electron vectors are multiplied by the "scale" factor to adjust for weaker or stronger deflection. After iterating over all electrons/vectors in the image and calculating their impact points on the virtual detector, the collected data is converted to a continuous probability density using the kdeplot function from the seaborn Python library, implementing a kernel density estimation. The result is a continuous grayscale image, where bright areas indicate areas towards which many electrons/vectors are deflected, whereas dark regions indicate regions with a below-average electron/vector incident count. The program additionally plots a representation of the spin structure as a discrete grid of arrows, whose length and direction indicate the local orientation and magnitude of the spin structure. The arrows' colors are mapped to their direction on a HSV spectrum using the arctan2 function from the Python library numpy and are shown in Fig. 1 b-f (panel III).” (line 638-668 and README-file at github.com/Julian-Hi/LTEM-contrast)

Reviewers' Comments:

Reviewer #2:

Remarks to the Author:

The authors have sufficiently responded to my previous questions and comments. As I said before, I believe the results are novel and deserve to be published. Thank you for your time editing the manuscript.